# Streaming Stochastic Submodular Maximization with On-Demand User Requests

**Honglian Wang**
KTH Royal Institute of Technology
Digital Futures
Stockholm, Sweden

**Sijing Tu**
KTH Royal Institute of Technology
Digital Futures
Stockholm, Sweden

**Lutz Oettershagen**
University of Liverpool
Liverpool, UK

**Aristides Gionis**
KTH Royal Institute of Technology
Digital Futures
Stockholm, Sweden

## Abstract

We explore a novel problem in streaming submodular maximization, inspired by the dynamics of news-recommendation platforms. We consider a setting where users can visit a news website at any time, and upon each visit, the website must display up to $k$ news items. User interactions are inherently stochastic: each news item presented to the user is consumed with a certain acceptance probability by the user, and each news item covers certain topics. Our goal is to design a streaming algorithm that maximizes the expected total topic coverage. To address this problem, we establish a connection to submodular maximization subject to a matroid constraint. We show that we can effectively adapt previous methods to address our problem when the number of user visits is known in advance or linear-size memory in the stream length is available. However, in more realistic scenarios where only an upper bound on the visits and sublinear memory is available, the algorithms fail to guarantee any bounded performance. To overcome these limitations, we introduce a new online streaming algorithm that achieves a competitive ratio of $1/(8\delta)$, where $\delta$ controls the approximation quality. Moreover, it requires only a single pass over the stream, and uses memory independent of the stream length. Empirically, our algorithms consistently outperform the baselines.

## 1 Introduction

Streaming submodular maximization has been extensively studied under various feasibility constraints, including *matchoid* [7, 8, 10], *matroid* [9], *knapsack* [21, 22] and *k-set* [20] constraints. In the classical setting, the goal is to select a single subset of items from a data stream and output it when the stream ends. However, this classical framework falls short in many real-world scenarios, for example, when seeking to maximize *content coverage* in online news recommendation. In such a setting, users visit a news website multiple times and expect to receive a relevant set of articles on each visit. Meanwhile, new articles continuously arrive at the content-provider's server in a stream fashion. The long-term objective is to maximize the total content coverage across user visits. Existing streaming submodular maximization algorithms are ill-suited for this use case, as they are designed to produce a single solution and do not support multiple or on-demand user interactions during the stream.

Motivated by this limitation, we introduce a novel problem called *Streaming Stochastic Submodular Maximization with On-demand Requests* (*S3MOR*). While the setting is broadly applicable to streaming submodular maximization tasks with multiple output requirements, we focus on the news

39th Conference on Neural Information Processing Systems (NeurIPS 2025).

Table 1: Comparison of our algorithms. $N \in \mathbb{N}$ denotes the stream length, $k \in \mathbb{N}$ number of news items to present per user access, $T \in \mathbb{N}$ and $T' \in \mathbb{N}$ are the exact number and upper bound of user accesses, and $\delta \in \mathbb{N}$ an approximation parameter.

| Algorithm | Competitive ratio | Time | Space | Response time |
|-----------|-------------------|------|-------|---------------|
| LMGREEDY | $1/2$ | $\mathcal{O}(NT'k)$ | $\Theta(N + kT)$ | $\mathcal{O}(Nk)$ |
| STORM | $1/4(T'-T+1)$ | $\mathcal{O}(NT'k)$ | $\mathcal{O}(T'k)$ | $\mathcal{O}(T')$ |
| STORM++ | $1/(8\delta)$ | $\mathcal{O}(NT'^2k/\delta)$ | $\mathcal{O}(T'^2k/\delta)$ | $\mathcal{O}(T')$ |

recommendation task for illustration. Therefore, in *S3MOR*, the data stream consists of incoming news items that the system aggregates into a personalized news website. Each item $V_i$ covers a set of topics and is associated with a probability $p_i$, representing the probability that a user will click on the item when it is presented. A user may visit the system arbitrarily often and at any time during the stream. At each visit, the system must select and present a subset of at most $k$ items from those that have arrived in the stream up to that point. If a user clicks on item $V_i$, they are exposed to all topics associated with that item. The objective is to maximize the expected total number of unique topics covered by all user clicks across all visits. This setting is challenging due to its online character:

- **On-demand**: At any point, the system must be ready to output a subset of news items.

- **Irrevocability**: Once a subset is presented to the user, it cannot be modified.

To the best of our knowledge, no existing algorithm directly addresses the *S3MOR* problem.

We evaluate the performance of online streaming algorithms that make irrevocable decisions using the notion of *competitive ratio*. An algorithm is called *c-competitive* if, for every input stream $\sigma$, the value $\mathcal{A}(\sigma)$ achieved by algorithm $\mathcal{A}$ satisfies $\frac{\mathcal{A}(\sigma)}{\mathbf{OPT}(\sigma)} \geq c$, where $\mathbf{OPT}(\sigma)$ denotes the value of an optimal offline solution with full access to the input stream.

To evaluate the performance of our algorithms, we clarify the notions of *response time complexity*, *time complexity*, and *space complexity*.

**Response-time complexity vs. time complexity.** For the *S3MOR* problem, we assume there exists an oracle that can evaluate the objective function $f(S)$ for any feasible candidate set $S$. We define the *response time complexity* of an algorithm as the worst-case number of oracle calls needed to produce a recommendation result of size $k$ after the user submits a request. This metric captures the latency users experience. In contrast, *time complexity* is defined as the worst-case number of oracle calls required by the algorithm over the entire stream.

**Space complexity.** In practice, all users share common storage for original news items. We distinguish between two storage types: *shared storage* containing original items accessible to all users (not counted toward space complexity), and *per-user storage* for additional memory each user needs to compute personalized recommendations, including document identifiers and user-item association probabilities. Throughout this paper, "space complexity" refers exclusively to per-user storage.

**Our contributions.** We reduce *S3MOR* to the problem of *streaming submodular maximization under a partition matroid constraint*, which allows us to leverage existing online streaming algorithms. Based on this reduction, we propose three algorithms and evaluate them across multiple dimensions: competitive ratio, space complexity, time complexity, and response time complexity. We state our main results as follows:

- We prove that provided sufficient memory to store the whole stream, a greedy algorithm, LMGREEDY, can achieve the best possible $1/2$ competitive ratio for the *S3MOR* problem.

- If the exact number $T$ of user visits is known, the reduction preserves the competitive ratio of the underlying online streaming submodular algorithm.

- However, in realistic scenarios, the number of user visits $T$ is unknown and we need to guess the number of visits as $T' > T$. For this case, we introduce an algorithm STORM, which has a competitive ratio of $\frac{1}{4(T'-T+1)}$.

- To achieve a better competitive ratio, we propose a second algorithm STORM++ that makes multiple concurrent guesses of $T$, maintaining one solution per guess. It then greedily

aggregates outcomes across guesses. This algorithm achieves a competitive ratio of $1/(8\delta)$, where $\delta$ is a tunable parameter that balances efficiency and solution quality. The space and time complexity increase by a factor of $\delta T'$ relative to the underlying streaming algorithm.

- STORM and STORM++ achieve worse competitive ratios than LMGREEDY, but they utilize much smaller space and have significantly better response time.

We summarize the results for all algorithms mentioned above in Table 1. We validate our approach through empirical experiments on both large-scale and small-scale real-world datasets. Our results show the effectiveness of our algorithms in terms of coverage quality and memory usage, in comparison to relevant baselines.

**Notation and background.** We assume familiarity with constrained submodular maximization and competitive analysis. We provide formal definitions in Appendix A.

### Related work

Submodular optimization plays a fundamental role in machine learning, combinatorial optimization, and data analysis [13, 26, 29], capturing problems with diminishing returns, and finding applications in data summarization [34, 42], non-parametric learning [17, 45], recommendation systems [1, 33, 38, 44], influence maximization [24], and network monitoring [13]. The classical greedy algorithm achieves a $(1 - 1/e)$-approximation for monotone submodular maximization under cardinality constraints [39], and extends to matroid constraints [6, 12].

We study stochastic submodular coverage functions, where the objective depends on random realizations but selections must be made non-adaptively. This contrasts with adaptive submodularity [16, 18, 37, 41], which models sequential decision-making with feedback and supports strong guarantees for greedy strategies. Related work includes stochastic variants of set and submodular cover, where the objective itself is random; notable examples analyze adaptivity gaps and approximation bounds under oracle access [15, 19].

Classical greedy algorithms for (constrained) submodular optimization [27, 35, 39] assume random access to the full dataset, which is infeasible in large-scale or streaming settings. This challenge has motivated streaming algorithms that make irrevocable decisions under memory constraints [4, 5, 9, 22, 23, 36]. Buchbinder et al. [5] apply the preemption paradigm, i.e., replacing elements with better ones, to achieve a tight $(1/2 - \varepsilon)$-approximation for monotone functions under cardinality constraints. Sieve++ [23] uses thresholding to achieve a $(1/2 - \varepsilon)$-approximation under cardinality constraints using only $\mathcal{O}(k)$ memory. Extensions to matroid and partition constraints include [7, 8, 10]; for instance, Chekuri et al. [8] gives a $1/4p$-competitive online streaming algorithm for monotone $p$-matchoids. These approaches, however, do not handle stochastic item utilities or repeated, unpredictable user access. We address this gap by studying stochastic submodular coverage under partition constraints in a streaming model with limited memory and unknown user access requests.

To extend submodularity to sequences, Alaei et al. [1] introduce sequence-submodularity and sequence-monotonicity, showing that a greedy algorithm achieves a $(1 - 1/e)$-approximation, with applications in online ad allocation. Similarly, Ohsaka and Yoshida [40] propose $k$-submodular functions to model optimization over $k$ disjoint subsets, providing constant-factor approximations for monotone cases. In contrast to their settings, our problem involves making irrevocable decisions as an incoming stream of items progresses, requiring each of the $k$ subsets to be constructed incrementally rather than all at once after seeing the entire input.

Finally, diversity-aware recommendation and coverage problems are closely related, as many systems must recommend item sets rather than single items, motivating submodular approaches to maximize coverage and user satisfaction [2, 43]. Ashkan et al. [3] propose a greedy offline method for maximizing utility under a submodular diversity constraint. Yue and Guestrin [44] use submodular bandits for diversified retrieval, learning user interaction probabilities adaptively rather than relying on a known oracle, as in our setting.

## 2 Streaming submodular maximization with on-demand requests

**Problem setting.** Let $\mathcal{V} = \{V_1, V_2, \ldots, V_N\}$ be an ordered set of $N$ news items appearing in the stream, where item $V_i$ arrives at time step $i$. Each news item $V_i \in \mathcal{V}$ covers a subset of topics from a

predefined set of topics $C = \{c_1, c_2, \ldots, c_d\}$, where $d$ denotes the total number of topics. Formally, $V_i \subseteq C$ for each $i \in [N]$. Additionally, each news item $V_i$ is associated with a probability[1] $p_i \in [0, 1]$, modeling the probability that a user will click the item when presented to them. When a user clicks a news item, we say that the user is exposed to all topics in the news item.

Moreover, we assume that we observe a binary variable $\tau_i \in \{0, 1\}$ indicating whether the user accesses the system at time step $i$. Specifically, $\tau_i = 1$ denotes an access at time step $i$, and $\tau_i = 0$ otherwise. Whenever $\tau_i = 1$, the system should present at most $k$ news items from the available news items $\{V_1, \ldots, V_i\}$. Now, suppose a user visits the system $T$ times. The system outputs $T$ sets $\mathcal{S}^1, \cdots, \mathcal{S}^T$, where $\mathcal{S}^t$ represents the items presented to the user at their $t$-th visit, for $t \in [T]$. We allow the same item to be presented at multiple visits; however, we treat each appearance as a distinct copy. We make this choice as the user has multiple chances to click on an item if the item is presented multiple times. Therefore, for any $t \neq s$, we have $\mathcal{S}^t \cap \mathcal{S}^s = \emptyset$. We define the set of all items presented to the user over the entire sequence of visits as $\mathcal{S} = \bigcup_{t=1}^{T} \mathcal{S}^t$.

**Objective function.** We define $f(\mathcal{S})$ as the expected number of topics the user covers after $T$ visits, where the expectation is taken over click probabilities. Let $\sigma_j \in \{0, 1\}$ indicate whether topic $c_j$ is covered, i.e., the user clicked at least one item that covers $c_j$ from the shown sets $\mathcal{S}^1, \cdots, \mathcal{S}^T$. The probability $\Pr(\sigma_j = 0)$ that topic $c_j$ remains uncovered equals the probability the user clicks no items covering $c_j$.

Since the user clicks on item $V_i$ with probability $p_i$, the probability that they do not click on it is $1 - p_i$ and, thus, $\Pr(\sigma_j = 0) = \prod_{t \in [T]} \prod_{V_i \in \mathcal{S}^t, V_i \ni c_j} (1 - p_i)$, leading to the expectation $\mathbb{E}[\sigma_j] = 1 - \prod_{t \in [T]} \prod_{V_i \in \mathcal{S}^t, V_i \ni c_j} (1 - p_i)$. By linearity of expectation, the expected number of topics covered by a user, which we denote by $f(\mathcal{S})$, is given by

$$f(\mathcal{S}) = \mathbb{E}\left[\sum_{j=1}^{d} \sigma_j\right] = \sum_{j=1}^{d} \mathbb{E}[\sigma_j] = \sum_{j=1}^{d} \left(1 - \prod_{t \in [T]} \prod_{V_i \in \mathcal{S}^t, V_i \ni c_j} (1 - p_i)\right). \tag{1}$$

**Lemma 2.1.** $f(\mathcal{S})$ *is a non-decreasing submodular set function.*

**Problem definition.** We formally define our new problem *streaming stochastic submodular maximization with on-demand requests (S3MOR)* below.

**Problem 2.2 (S3MOR).** *We define a news item stream* **S** *as a sequence of triples:* $\mathbf{S} = ((V_1, p_1, \tau_1), \ldots, (V_N, p_N, \tau_N))$ *where $N$ is the (unknown) length of the stream. At each time step $i$, the system receives a triple $(V_i, p_i, \tau_i)$, where $V_i \in \mathcal{V}$, $p_i \in [0, 1]$, and $\tau_i \in \{0, 1\}$. Whenever $\tau_i = 1$, the system selects up to $k$ news items from the set of items $\{V_1, \ldots, V_i\}$ received so far to present to the user. Let $\mathcal{S} = \{\mathcal{S}^1, \ldots, \mathcal{S}^T\}$ denote the sets of news items presented to the user over $T$ accesses. The objective is to maximize the expected number of distinct topics the user is exposed to by the end of the stream, measured by the function $f(\mathcal{S})$.*

There are two key challenges in our streaming setting: (i) the system does not know in advance when or how many times a user will access it; and (ii) the system has to make decisions based on the information available so far, and these decisions are irrevocable. In the following, we show how the problem can be reduced to a submodular maximization problem subject to a partition matroid constraint. We then leverage this connection to develop solutions under various memory constraints.

**Reduction to submodular maximization under a partition matroid constraint.** We first reduce the *S3MOR* problem (Problem 2.2) to submodular maximization under a partition matroid constraint, which enables us to apply efficient streaming algorithms with provable guarantees. Let $r : t \mapsto r_t$ map a user's $t$-th access to its access time, and $(r_t)_{t=1}^{T}$ be the sequence of times at which the user accesses the system over an unknown time period. Whenever the user accesses the system, it presents up to $k$ news items selected from all the news items available since the beginning of the stream, denoted by $\{V_i\}_{i=1}^{r_t}$.

---

[1]These probabilities can be interpreted as measures of user-item preference or engagement, and are typically estimated from historical click-through data using machine learning models such as logistic regression, factorization machines, or neural networks trained on user behavior patterns and item features. We assume they are static and remain constant over time.

Let $V_i^t$ be a *copy* of the item $V_i$ associated with the $t$-th visit. Let $\mathcal{V}^t$ be a collection of these copies until the $t$-th visit, namely, $\mathcal{V}^t = \{V_i^t\}_{i=1}^{r_t} = \{V_1^t, \ldots, V_{r_t}^t\}$. Without loss of generality, we treat each copy as a distinct item, i.e., $V_i^t \neq V_i^s$ for all $t \neq s$. Clearly, $\mathcal{V}^t \cap \mathcal{V}^s = \emptyset$ for $t \neq s$.

Moreover, let $\mathcal{S}^t \subseteq \mathcal{V}^t$ be the set of news items that the system presents to the user at their $t$-th visit. Let $\mathcal{S} = \bigcup_{q=1}^T \mathcal{S}^q$ be the union of all sets of presented items. Analogously to Equation (1), we adapt the function $f\colon 2^{\cup_{t=1}^T \mathcal{V}^t} \to \mathbb{R}$ such that $f(\mathcal{S})$ is the expected number of topics covered by the (copies) of news items that the user clicked on after $\mathcal{S}^1, \ldots, \mathcal{S}^T$ have been presented to the user.

**Observation 2.3.** *Let $(\cup_{t=1}^T \mathcal{V}^t, \mathcal{F})$ be a set system, where $\mathcal{F}$ is a collection of subsets of $\cup_{t=1}^T \mathcal{V}^t$, and $\mathcal{F}$ is constructed in the following way: for any set $X \subseteq \cup_{t=1}^T \mathcal{V}^t$, if $|X \cap \mathcal{V}^t| \leq k$ for any $1 \leq t \leq T$, then $X \in \mathcal{F}$. Observe that $(\cup_{t=1}^T \mathcal{V}^t, \mathcal{F})$ is a partition matroid. Problem 2.2 can be equivalently formulated as $\max_{X \in \mathcal{F}} f(X)$.*

Based on Observation 2.3, we develop a baseline algorithm for the Problem 2.2. Suppose that the system stores all the news items up to when the user accesses the system at the $t$-th time, and also has stored all the previous news item sets presented to the user, i.e., $\cup_{q=1}^{t-1} \mathcal{S}^q$. In this case, the stored news items up to time $r_t$ constitute one partition of the matroid, i.e., $\mathcal{V}^t = \{V_i^t\}_{i=1}^{r_t}$. We then apply the local greedy algorithm by Fisher et al. [12], i.e., the system greedily selects $k$ news items from $\mathcal{V}^t$ that maximize the marginal gain of the objective function with respect to $f(\cup_{q=1}^{t-1} \mathcal{S}^q)$. This algorithm does not require any knowledge of $T$, and has a competitive ratio of $1/2$. Since all incoming news items need to be stored, the memory usage is $\Omega(N + kT)$, and the running time is $\sum_{q=1}^T \mathcal{O}(|\mathcal{V}^q|k)$. The response time is $\mathcal{O}(Nk)$ since it selects $k$ items iteratively from $\mathcal{O}(N)$ items. We present the algorithm in Algorithm 3 (LMGREEDY) in Appendix B.

**Theorem 2.4.** *Algorithm 3 for Problem 2.2 has a competitive ratio of $1/2$, space complexity $\Theta(N + kT)$, time complexity $\mathcal{O}(NTk)$, and response time complexity $\mathcal{O}(Nk)$. Moreover, the competitive ratio of $1/2$ is tight, i.e., for Problem 2.2, no streaming algorithm can achieve a competitive ratio better than $1/2$ without violating the irrevocability constraint.*

The drawback of LMGREEDY is that its memory usage depends linearly on the stream length, making it infeasible for large streams. Thus, we consider the setting where the system has limited memory and can store only a user-specific subset of news items.

## 3 Algorithms with limited memory

We now consider a more realistic setting where the system has limited memory. Our objective is to substantially reduce the memory usage and the response time complexity while achieving a bounded approximation ratio. We first establish that any algorithm achieving reasonably good performance must use at least $\Theta(Tk)$ memory.

To illustrate this lower bound, consider the following example: Suppose that a user makes $T$ requests; the input stream contains more than $Tk$ items, each containing one distinct topic and associated with acceptance probability of 1. At the end of the stream, the user sequentially submits $T$ requests. If an algorithm is allocated with only $\Theta(T'k)$ memory for some $T' \leq T$, it can store at most $\Theta(T'k)$ items, thus, at most $\Theta(T'k)$ distinct topics. However, optimally, the user may access $\Theta(Tk)$ distinct topics. We can conclude that for this example, the competitive ratio is at most $\Theta(T'/T)$, which is arbitrarily poor when $T' \ll T$.

Since the exact number of requests $T$ is typically unknown in advance, we assume the system has access to an upper bound $T' \geq T$ and is allocated $\Omega(kT')$ memory. This allocation is sufficient to store all candidate items throughout the user's interaction period. In practice, $T'$ can be estimated from historical interaction data, allowing the system to set $T'$ close to $T$ within reasonable accuracy, ensuring both efficient memory utilization and meaningful performance guarantees.

### 3.1 A first memory-efficient streaming algorithm

Our first memory-efficient algorithm processes the incoming stream **S** as follows. First, it initializes $T'$ empty candidate sets $\mathcal{A}^i$ for $i \in [T']$, all of which are initially *active*. Upon the arrival of a news item from the stream **S**, for each active candidate set, the algorithm either adds the item to the set or

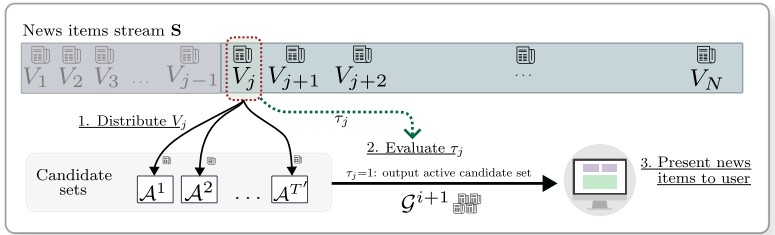

Figure 1: Schematic view of our algorithm STORM. We first initialize $T'$ empty active candidate sets. For each incoming news item in the stream $\mathbf{S}$, we decide whether it can be added to/swapped into each of the active candidate sets. When a user submits a request, i.e., $\tau_j = 1$, we select the best active candidate set, present it to the user, and deactivate it.

---

**Algorithm 1:** STORM

**Input:** Estimated visits $T' \in \mathbb{N}$, budget $k \in \mathbb{N}$.

1   $\mathcal{A}^1, \cdots, \mathcal{A}^{T'} \leftarrow \emptyset, \mathcal{T} = \{1, \cdots, T'\}, \mathcal{G}^0 \leftarrow \emptyset, i \leftarrow 0$

2   **for** $(V, \tau, p)$ *in the stream* **do**

3      **for** $j \in \mathcal{T}$ **do**

4         **if** $|\mathcal{A}^j| < k$ **then**

5            $\mathcal{A}^j \leftarrow \mathcal{A}^j \cup \{V\}$

6         **else**

7            $V' \leftarrow \arg\min_{V \in \mathcal{A}^j} \nu(f, \cup_{q \in [T']}\mathcal{A}^q, V)$

8            **if** $f(V \mid \cup_{q \in [T']}\mathcal{A}^q) \geq 2\nu(f, \cup_{q \in [T']}\mathcal{A}^q, V')$ **then**

9              $\mathcal{A}^j \leftarrow \mathcal{A}^j \setminus \{V'\} \cup \{V\}$

10     **if** $\tau = 1$ **then**

11       $j^* \leftarrow \arg\max_{j \in \mathcal{T}'} f(\mathcal{A}^j | \bigcup_{t=0}^{i} \mathcal{G}^t)$

12       $\mathcal{T} \leftarrow \mathcal{T} \setminus \{j^*\}$                 ▷ Deactivate candidate set $\mathcal{A}^{j^*}$

13       $\mathcal{G}^{i+1} \leftarrow \mathcal{A}^{j^*}$

14       **output** $\mathcal{G}^{i+1}$

15       $i \leftarrow i + 1$

---

replaces an existing item if the set is full. When a user accesses the system, the algorithm selects one of the active candidate sets to present to the user. The chosen candidate set is then marked as *inactivate* and will not be updated further. Figure 1 shows an overview.

Based on our reduction to submodular maximization under a partition matroid constraint, we can adapt any streaming algorithm designed for matroid constraints, provided it produces irrevocable outputs. Notable examples include the algorithms proposed by Chakrabarti and Kale [7], Chekuri et al. [8], and Feldman et al. [10], all of which achieve a $1/4$ approximation ratio. In our work, we specifically adapt the algorithm introduced by Chekuri et al. [8].

Our algorithm, named STORM, is shown in Algorithm 1. It keeps track of the number of visits using a counter $i$. Assume the user has visited $i$ times when news item $V_j$ arrives. This implies that, prior to the arrival of $V_j$, $i$ active sets from $i$ partitions have already been presented to the user, and thus, neither $V_j$ nor any subsequent item will be added to these partitions. The algorithm iterates over the remaining $T' - i$ active candidate sets, and determines whether $V_j$ should be added to any of them, with replacement if necessary.

The decision to replace an existing item $V$ with the current item $V_j$ is based on the comparison of their incremental value with respect to set $\cup_{t \in [T']}\mathcal{A}^t$. Specifically, we adopt the notation $\nu(f, \cup_{t \in [T']}\mathcal{A}^t, V)$ from Chekuri et al. [8] and define

$$\nu(f, \cup_{t \in [T']}\mathcal{A}^t, V) = f_{\widehat{\cup_{t \in [T']}\mathcal{A}^t}}(V), \text{ with } \widehat{\cup_{t \in [T']}\mathcal{A}^t} = \{V' \in \cup_{t \in [T']}\mathcal{A}^t : V' < V\}.$$

---

**Algorithm 2:** STORM++

---

**Input:** Estimated visits $T' \in \mathbb{N}$, budget $k \in \mathbb{N}$, parameter $\delta \in \mathbb{N}$.

1   $\mathcal{G}^0 \leftarrow \emptyset, \mathcal{B}^0 \leftarrow \emptyset, i \leftarrow 0, \mathcal{P} \leftarrow \left\{ \delta, 2\delta, \ldots, \lceil \frac{T'}{\delta} \rceil \delta \right\}$

2   **for** $\tilde{T} \in \mathcal{P}$ **do** Initialize STORM $(\tilde{T}, k)$            $\triangleright$ Maintain STORM for different $\tilde{T}$

3   **for** $(V, \tau, p)$ *in the stream* **do**

4      **for** $\tilde{T} \in \mathcal{P}$ **do**

5          $\mathcal{G}^{i+1,(\tilde{T})} \leftarrow$ STORM $(\tilde{T}, k).\texttt{step}(V, \tau, p)$    $\triangleright$ Process $(V, \tau, p)$ by executing lines 3-15 of Alg 1

6      **if** $\tau = 1$ **then**

7          $\tilde{T}^* \leftarrow \arg\max_{\tilde{T} \in \mathcal{P}} f(\mathcal{G}^{i+1,(\tilde{T})} \mid \bigcup_{j=0}^{i} \mathcal{B}^j)$

8          $\mathcal{B}^{i+1} \leftarrow \mathcal{G}^{i+1,(\tilde{T}^*)}$

9          **output** $\mathcal{B}^{i+1}$

10         $i \leftarrow i + 1$

---

Here, $V' < V$ indicates item $V'$ is added to the set $\cup_{t \in [T']} \mathcal{A}^t$ before item $V$. For a specific set $\mathcal{A}^{t'}$, $V_j$ replaces an item in $\mathcal{A}^{t'}$ if $f_{\cup_{t \in [T']} \mathcal{A}^t}(V_j) \geq 2 \cdot \min_{V \in \mathcal{A}^{t'}} \nu(f, \cup_{t \in [T']} \mathcal{A}^t, V)$.

Upon the $t$-th user visit, the system selects which active candidate set to output in a greedy manner. To be specific, let $\mathcal{G}^1, \cdots, \mathcal{G}^{t-1}$ denote the candidate sets that have already been presented to the user, and $\mathcal{T}$ represent the indices of the remaining active sets. The system selects the active set that offers the highest incremental gain with respect to the union of previously presented sets. Formally, let $j^* = \arg\max_{j \in \mathcal{T}} f(\mathcal{A}^j \mid \cup_{j=1}^{t-1} \mathcal{G}^j)$, and the system outputs $\mathcal{G}^t = \mathcal{A}^{j^*}$.

**Theorem 3.1.** *Let $T$ be the number of user accesses and $T'$ a given upper bound. Algorithm 1 has competitive ratio $\frac{1}{4(T'-T+1)}$, space complexity $\mathcal{O}(T'k)$, time complexity $\mathcal{O}(NkT')$, and response time complexity $\mathcal{O}(T')$.*

If the exact number of visits $T$ is known, we can set the upper bound $T' = T$ and obtain a $1/4$-competitive ratio using Algorithm 1. However, the competitive ratio of Algorithm 1 can be arbitrarily bad if $T'$ is significantly larger than $T$. Note that our analysis is tight. We can demonstrate that for any fixed $T$, there exists an example such that the competitive ratio of Algorithm 1 is at most $\frac{1}{T'-T+1}$, which is only a constant factor away from our analysis. We provide the proof in Appendix C.

### 3.2 A memory-efficient streaming algorithm with bounded competitive ratio

As shown in the previous section, to achieve a large enough competitive ratio with STORM, the system must have a good enough estimate of the number of user visits $T$. However, in practice, this information is typically unavailable in advance. To address this issue, we discretize the range $[T']$ to generate multiple guesses for the value of $T$, and run a separate instance of Algorithm 1 for each guess. When a user visits the system, each copy of Algorithm 1 outputs a solution, and we select the best solution over all the outputs.

Specifically, Algorithm 2 (named STORM++) first initializes the set of guesses of $T$ as $\mathcal{P} = \{\delta i \mid i \in [\lceil T'/\delta \rceil]\}$, where $\delta \in \mathbb{N}$ is a parameter controlling the trade-off between competitive ratio as well as the space and time complexity. In this construction, there always exists a guess $\tilde{T} \in \mathcal{P}$ that is between $T$ and $T + \delta$. Therefore, if we would run Algorithm 1 with $\tilde{T}$ as the input parameter, and present the results to the user, we are guaranteed to have a competitive ratio of at least $1/4\delta$ as shown in Theorem 3.1; however, we do not know the value of $\tilde{T}$ in advance. Thus, we adopt a greedy strategy to aggregate the results from all the copies of Algorithm 1 to determine an output for each user visit. At the $i$-th user visit, we collect the $i$-th output from each running copy of STORM with a different guess $\tilde{T} \in \mathcal{P}$ and select the one that maximizes the incremental gain. This selection rule leads to the following result.

**Theorem 3.2.** *Algorithm 2 has a competitive ratio of $1/(8\delta)$, space complexity $\mathcal{O}(T'^2 k/\delta)$, time complexity $\mathcal{O}(NkT'^2/\delta)$, and response time complexity $\mathcal{O}(T')$.*

# 4 Empirical evaluation

We empirically evaluate the performance of our proposed algorithms STORM and STORM++. We denote $\text{STORM}(T)$ and $\text{STORM}(T')$ as the STORM algorithm with the input parameter set to $T$ (i.e., exact number of user accesses) and $T'$ (i.e., upper bound of user accesses), resp.

We design our experiments to answer the following research questions: **(Q1)** How effective are the solutions provided by STORM++ and $\text{STORM}(T')$ compared to other baselines and to each other? **(Q2)** What is the impact of parameters $\delta$ and $k$ on the performance of STORM++ and $\text{STORM}(T')$? **(Q3)** How sensitive are STORM++ and $\text{STORM}(T')$ to the upper bound $T'$ and do our algorithms require highly-accurate estimation of $T'$ to perform well?

We use the following **baselines**:

- LMGREEDY: Our upper-bound baseline, which assumes linear memory size in the stream length, i.e., potentially unbounded memory (Algorithm 3).

- SIEVE++: Based on Kazemi et al. [23], we implement a heuristic version of the SIEVE++ algorithm to solve the *S3MOR* problem. The adaptation proceeds as follows: (1) Before the first user visit, we run the exact SIEVE++ algorithm and obtain output set $\mathcal{S}^1$. (2) For each subsequent $i$-th output, we run SIEVE++ on the data stream between the $(i-1)$-th and $i$-th user visits, using the marginal gain function $f(\cdot|\bigcup_{j=1}^{i-1}\mathcal{S}^i)$ as the objective.

- PREEMPTION: We adapt the algorithm proposed by Buchbinder et al. [5] for solving the *S3MOR* problem in the same way as described above for SIEVE++.

**Datasets:** We use six real-world datasets, including four rating datasets KuaiRec, Anime, Beer and Yahoo, and two multi-label dataset RCV1 and Amazon. The number of items these dataset contains vary from thousands to one million, and the topic sets size vary from around 40 to around 4000. We provide the details in Appendix D.1.

**Experimental setting:** To simulate the news item stream, we randomly shuffle the items to form a stream of length $N$. To simulate the user visit sequence, we start with an all-zeros sequence and then randomly select $T$ indices, and set those positions to one. We establish an upper bound on the number of visits by setting $T' = T + \Delta T$ for some positive $\Delta T$. We generate synthetic click probabilities, and provide the details in Appendix D.1.

In all experiments, we randomly selected 50 users and report the average expected coverage for the Yahoo, RCV1, and Amazon datasets. Standard deviations and expected coverage results for the other three datasets are provided in Appendix D. We also report runtime and memory usage on the largest dataset, Amazon.

All experiments ran on a MacBook Air (Model Mac15,12) equipped with an Apple M3 chip (8-core: 4 performance and 4 efficiency cores), 16 GB of memory, and a solid-state drive (SSD). The source code and datasets are available.[2]

**Results and discussion**

**Impact of $k$ and $T$ on the expected coverage.** As shown in Figure 2, LMGREEDY achieves the best performance across all datasets and values of $k$. Among our proposed methods, $\text{STORM}(T)$ consistently performs the best, followed by STORM++, then $\text{STORM}(T')$. Compared to baselines SIEVE++ and PREEMPTION, our algorithms generally perform better across datasets and values of $k$, with exceptions on RCV1, where PREEMPTION performs comparably to $\text{STORM}(T')$. In Figure 3, we observe a trend shift: STORM++ slightly outperforms $\text{STORM}(T')$ and $\text{STORM}(T)$ across datasets and $T$ values. $\text{STORM}(T')$ and $\text{STORM}(T)$ perform comparably. This performance shift is due to smaller $\delta$ and $\Delta T$ in this experiment, which lead to improved competitive ratios for $\text{STORM}(T')$ and STORM++. Although $\text{STORM}(T)$ offers the strongest theoretical guarantee of the three algorithms, $\text{STORM}(T')$ and STORM++ can empirically outperform it (we give an example of this effect in Appendix D.2).

**Performance regarding parameters $\delta$ and $\Delta T$.** Figure 4a shows that STORM++'s expected coverage decreases as $\delta$ increases, consistent with theory: as $\delta$ grows from 3 to 30, the performance guarantee

---

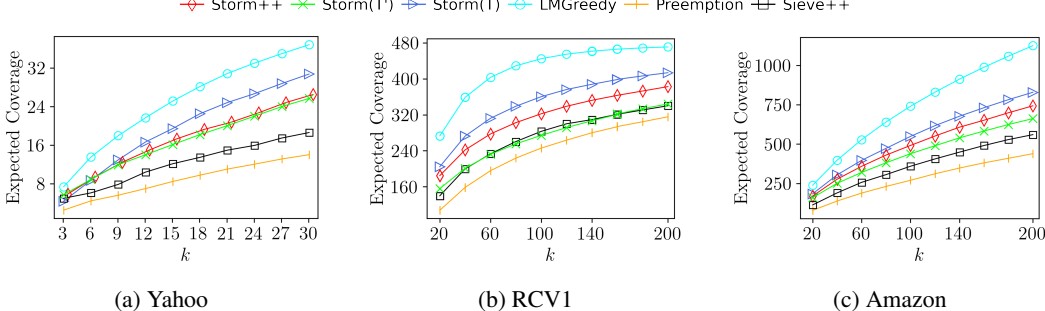

Figure 2: Empirical variation in expected coverage as a function of budget $k$ on all datasets. Parameter $T = 5$, $\delta = 25$ and $\Delta T = 45$ are fixed

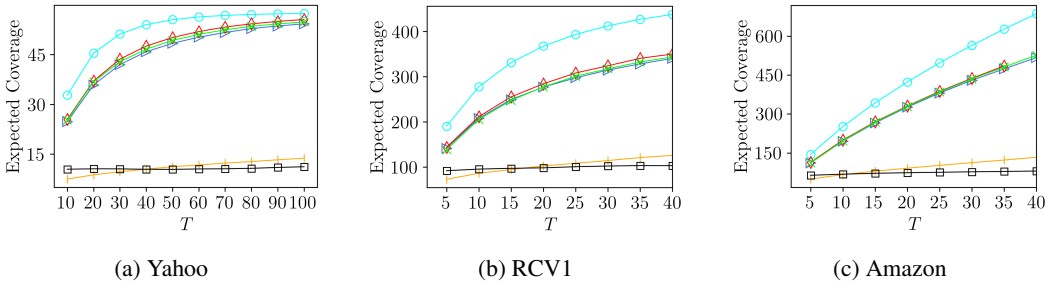

Figure 3: Empirical variation in expected coverage as a function of number of visits $T$ on all datasets. Parameter $k = 10$, $\delta = 10$ and $\Delta T = 10$ are fixed.

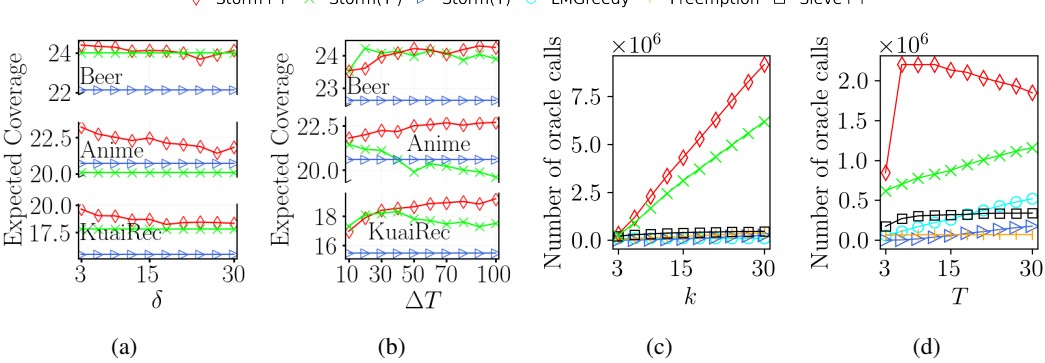

Figure 4: Left two: Impact of $\Delta T$ and $\delta$ on expected coverage. In (a) and (b), we fix $k = 5$ and $T = 10$. When varying $\delta$, we fix $\Delta T = 60$; when varying $\Delta T$, we fix $\delta = 10$. Right two: Oracle call counts on the Beer dataset. We fix $\delta = 25$ and $\Delta T = 45$. When varying $k$, we fix $T = 5$; when varying $T$, we fix $k = 5$.

Table 2: Runtime and mem. usage for the Amazon dataset. We set $k$, $T$, and $\delta$ to 10, $T' = 20$.

|  | STORM++ | STORM($T'$) | STORM($T$) | LMGREEDY | SIEVE++ | PREEMPTION |
|---|---|---|---|---|---|---|
| Runtime (s) | 35.13 | 7.94 | 4.38 | 88.13 | 307.43 | 136.04 |
| Memory (MB) | 2.00 | 1.28 | 0.72 | 28.99 | 0.91 | 0.03 |

drops by a factor of 10. This is reflected empirically with coverage declines across datasets, for example, from 130 to 120 on Amazon.

Figure 4b shows that STORM($T'$)'s performance declines with increasing $\Delta T$, matching the theoretical drop in competitive ratio (e.g., 10-fold decrease from $\Delta T = 10$ to 100). In contrast, STORM++ is theoretically unaffected.

**Number of oracle calls, runtime and memory.** Figure 4c shows that the number of oracle calls increases rapidly and linearly with budget $k$ for both STORM($T'$) and STORM++, while remaining

relatively stable for other algorithms. Similarly, Figure 4d indicates a linear increase in oracle calls with user visits $T$ for STORM($T'$), STORM($T$), and LMGREEDY, but a much slower growth for SIEVE++ and PREEMPTION. Notably, Figure 4d also reveals a non-monotonic trend: oracle calls first rise, then decline with increasing $T$. This increase is due to the guessing strategy for $T$ under $\delta = 25$ and $\Delta T' = 45$, where the number of guesses $\mathcal{T}$ increases ($\mathcal{T} = \{25, 50\}$ for $T = 3$ and $\mathcal{T} = \{25, 50, 75\}$ for $T \in [6, 30]$). The decrease is due to STORM++'s irrevocability—as more subsets are presented to the user, fewer items remain in $\mathcal{S}$ for potential swap, thereby reducing the number of subsequent oracle calls.

Table 2 shows that our proposed algorithms require much less memory than LMGREEDY, and use comparable memory to the other two baselines. STORM($T'$) and STORM($T$) are also significantly faster than the remaining algorithms.

## 5   Conclusion

We investigate a novel online setting for streaming submodular maximization where users submit on-demand requests throughout the stream. In this context, we introduce the *Streaming Stochastic Submodular Maximization with On-demand Requests* (*S3MOR*) problem with the goal of maximizing the expected coverage of the selected items. We first propose a memory-efficient approximation algorithm that highlights the trade-off between memory usage and solution quality, but its competitive ratio can be poor. To solve this, we propose a parameter-dependent memory-efficient approximation algorithm to trade off between the competitive ratio and the memory and time usage. Our empirical evaluations show that our proposed algorithms consistently outperform the baselines. We believe our approach will have a positive social impact. For instance, it can be applied to enhance the diversity of news content recommended to the users, thereby helping to expose them to multiple viewpoints and broaden their knowledge. While our approach optimizes topic coverage and can be used to promote content diversity, its misuse, such as deliberately optimizing a submodular utility function that favors polarized or biased content, could risk amplifying existing biases. However, such outcomes would require intentional manipulation of the objective function. When applied responsibly, our approach does not pose such risks.

## Acknowledgments and Disclosure of Funding

This research is supported by the ERC Advanced Grant REBOUND (834862), the Swedish Research Council project ExCLUS (2024-05603), and the Wallenberg AI, Autonomous Systems and Software Program (WASP) funded by the Knut and Alice Wallenberg Foundation.

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

# Appendix

## A  Preliminaries

We use $[j]$, with $j \in \mathbb{N}$, to denote the set $\{1, 2, \ldots, j\}$. A *set system* is a pair $(U, \mathcal{I})$, where $U$ is a ground set and $\mathcal{I}$ is a family of subsets of $U$.

**Definition A.1** (Matroid). *A set system $(U, \mathcal{I})$ is a* matroid *if*

  (i)  $\emptyset \in \mathcal{I}$,

  (ii)  *for $B \in \mathcal{I}$ and any $A \subseteq B$ we have $A \in \mathcal{I}$, and*

  (iii)  *if $A \in \mathcal{I}$, $B \in \mathcal{I}$ and $|A| < |B|$, then there exists $e \in B \setminus A$ such that $A \cup \{e\} \in \mathcal{I}$.*

**Definition A.2** (Partition matroid). *A matroid $\mathcal{M} = (U, \mathcal{I})$ is a* partition matroid *if there exists a partition $\{E_i\}_{i=1}^{r}$ of $U$, i.e., $U = \cup_{i=1}^{r} E_i$ and $E_i \cap E_j = \emptyset$ for all $i \neq j$, and non-negative integers $k_1, \ldots, k_r$, such that $\mathcal{I} = \{I \subseteq U : |I \cap E_i| \leq k_i \text{ for all } i \in [r]\}$.*

A non-negative set function $f : 2^U \to \mathbb{R}_{\geq 0}$ is called *submodular* if for all sets $A \subseteq B \subseteq U$ and all elements $e \in U \setminus B$, it is $f(A \cup \{e\}) - f(A) \geq f(B \cup \{e\}) - f(B)$. The function $f$ is *monotone* if for all $A \subseteq B$ it is $f(A) \leq f(B)$. Let $U$ be a finite ground set, and let $f : 2^U \to \mathbb{R}_{\geq 0}$ be a *monotone submodular* set function. The *constrained submodular maximization problem* is to find $\max_{S \in \mathcal{I}} f(S)$, where $\mathcal{I} \subseteq 2^U$ represents the feasibility constraints.

For ease of notation, for any two sets $A \subseteq U$ and $B \subseteq U$, and any item $e \in U$, we write $A + B$ for $A \cup B$, $A + e$ for $A \cup \{e\}$, $A - B$ for $A \setminus B$, and $A - e$ for $A \setminus \{e\}$. We write $f(A \mid B)$ and $f(e \mid B)$ for the incremental gain respectively defined as $f(A \cup B) - f(B)$ and $f(A \cup \{e\}) - f(A)$.

## B  Omitted proofs of Section 2

**Lemma 2.1.** $f(\mathcal{S})$ *is a non-decreasing submodular set function.*

*Proof.* For any set $\mathcal{S}$, we can rewrite Equation (1). Denote $\mathcal{S} = \cup_{t=1}^{T} \mathcal{S}^t$ and we allow multiple copies of $V_i$ to appear in $\mathcal{S}$ (the copies come from each $\mathcal{S}^t$). Note that $\mathcal{S}$ is a multi-set, and

$$f(\mathcal{S}) = \sum_{j=1}^{d} \left( 1 - \prod_{t \in [T]} \prod_{V_i \in \mathcal{S}^t, V_i \ni c_j} (1 - p_i) \right) = \sum_{j=1}^{d} \left( 1 - \prod_{V_i \in \mathcal{S}, V_i \ni c_j} (1 - p_i) \right).$$

Let $g_j(\mathcal{S}) = 1 - \prod_{V_i \in \mathcal{S}, V_i \ni c_j} (1 - p_i)$. Then $f(\mathcal{S}) = \sum_{j=1}^{d} g_j(\mathcal{S})$. To prove the lemma, it suffices to show that $g_j$ is non-decreasing and submodular for all $j \in [d]$.

**Non-decreasing property**: For any two sets $\mathcal{X} \subseteq \mathcal{Y}$ and for all $j \in [d]$, it is

$$g_j(\mathcal{Y}) - g_j(\mathcal{X}) = \left( 1 - \prod_{V_i \in \mathcal{Y} \setminus \mathcal{X}, V_i \ni c_j} (1 - p_i) \right) \prod_{V_i \in \mathcal{X}, V_i \ni c_j} (1 - p_i) \geq 0.$$

**Submodularity**: Let $\mathcal{X} \subseteq \mathcal{Y}$, and let $Z \in \overline{\mathcal{Y}}$ be an element not in $\mathcal{Y}$. Then, for all $j \in [d]$, the following holds:

$$g_j(\mathcal{Y} \cup \{Z\}) - g_j(\mathcal{Y}) - [g_j(\mathcal{X} \cup \{Z\}) - g_j(\mathcal{X})]$$

$$= \left( 1 - \prod_{V_i \in \{Z\}, V_i \ni c_j} (1 - p_i) \right) \prod_{V_i \in \mathcal{Y}, V_i \ni c_j} (1 - p_i) - \left( 1 - \prod_{V_i \in \{Z\}, V_i \ni c_j} (1 - p_i) \right) \prod_{V_i \in \mathcal{X}, V_i \ni c_j} (1 - p_i)$$

$$= \left( 1 - \prod_{V_i \in \{Z\}, V_i \ni c_j} (1 - p_i) \right) \left( \prod_{V_i \in \mathcal{Y} \setminus \mathcal{X}, V_i \ni c_j} (1 - p_i) - 1 \right) \prod_{V_i \in \mathcal{X}, V_i \ni c_j} (1 - p_i) \leq 0.$$

$\square$

---

**Algorithm 3:** LMGREEDY

---

**Input:** Budget $k \in \mathbb{N}$.

1 $\mathcal{G}^0, \cdots, \mathcal{G}^T \leftarrow \emptyset, \mathcal{G} \leftarrow \emptyset, \mathcal{V}^1 \leftarrow \emptyset, t = 1$

2 **for** $(V, \tau, p)$ *in the stream* **do**

3    $\mathcal{V}^t \leftarrow \mathcal{V}^t \cup \{V\}$

4    **if** $\tau = 1$ **then**

5       $\mathcal{N}^t \leftarrow \mathcal{V}^t$

6       **for** $j = 1$ *to* $k$ **do**

7          $V^* \leftarrow \arg\max_{V \in \mathcal{N}^t} f(\mathcal{G} \cup \mathcal{G}^t \cup \{V\})$

8          $\mathcal{G}^t \leftarrow \mathcal{G}^t \cup \{V^*\}$

9          $\mathcal{N}^t \leftarrow \mathcal{N}^t \setminus V^*$

10       **output** $\mathcal{G}^t$

11       $\mathcal{G} = \mathcal{G} \cup \mathcal{G}^t$

12       $\mathcal{V}^{t+1} \leftarrow \mathcal{V}^t$

13       $t \leftarrow t + 1$

---

**Theorem 2.4.** *Algorithm 3 for Problem 2.2 has a competitive ratio of* $1/2$, *space complexity* $\Theta(N + kT)$, *time complexity* $\mathcal{O}(NTk)$, *and response time complexity* $\mathcal{O}(Nk)$. *Moreover, the competitive ratio of* $1/2$ *is tight, i.e., for Problem 2.2, no streaming algorithm can achieve a competitive ratio better than* $1/2$ *without violating the irrevocability constraint.*

*Proof.* We first prove that Algorithm 3 is $1/2$ competitive for Problem 2.2.

Let $\mathcal{O}^1, \cdots, \mathcal{O}^T$ be the optimal output sets, and let $\mathcal{G}^1, \cdots, \mathcal{G}^T$ be the output sets obtained by Algorithm 3. Since we treat each copy of the same item as a distinct item, it follows that for any $i \neq j$, $\mathcal{O}^i \cap \mathcal{O}^j = \emptyset$ and $\mathcal{G}^i \cap \mathcal{G}^j = \emptyset$. We let $\mathcal{G} = \bigcup_{t=1}^T \mathcal{G}^t$ and $\mathcal{O} = \bigcup_{t=1}^T \mathcal{O}^t$.

We prove that the following holds:

$$f(\mathcal{O}) \overset{(a)}{\leq} f(\mathcal{G}) + \sum_{V \in \mathcal{O} \setminus \mathcal{G}} f(V \mid \mathcal{G}) \overset{(b)}{=} f(\mathcal{G}) + \sum_{i=1}^T \sum_{V \in \mathcal{O}^i \setminus \mathcal{G}} f(V \mid \mathcal{G}) \tag{2}$$

$$\overset{(c)}{\leq} f(\mathcal{G}) + \sum_{t=1}^T \sum_{V \in \mathcal{O}^i} f(V \mid \mathcal{G}) \overset{(d)}{\leq} f(\mathcal{G}) + \sum_{t=1}^T \sum_{V \in \mathcal{O}^i} f(V \mid \bigcup_{j=1}^i \mathcal{G}^j).$$

Inequalities $(a)$ and $(d)$ holds by submodularity, $(c)$ holds by monotonicity, and equality $(b)$ holds because $\{\mathcal{O}^1, \cdots, \mathcal{O}^T\}$ form a partition of $\mathcal{O}$.

Next, we bound the final term of Equation (2). We denote $\mathcal{O}^i = \{O_1^i, \cdots, O_k^i\}$, and $\mathcal{G}^i = \{G_1^i, \cdots, G_k^i\}$, with $G_j^i$ representing the $j$-th item that is added to $\mathcal{G}^i$ during the greedy selection steps. Note that, we can assume $|\mathcal{G}| = |\mathcal{O}| = k$, because $f$ is monotone. The following holds

$$\sum_{V \in \mathcal{O}^i} f(V \mid \bigcup_{j=1}^i \mathcal{G}^j) = \sum_{l=1}^k f(O_l^i \mid \bigcup_{j=1}^i \mathcal{G}^j) \overset{(a)}{\leq} \sum_{l=1}^k f(O_l^i \mid \bigcup_{j=1}^{i-1} \mathcal{G}^j + \bigcup_{s=1}^{l-1} G_s^i) \tag{3}$$

$$\overset{(b)}{\leq} \sum_{l=1}^k f(G_l^i \mid \bigcup_{j=1}^{i-1} \mathcal{G}^j + \bigcup_{s=1}^{l-1} G_s^i) = f(\bigcup_{j=1}^i \mathcal{G}^j - \bigcup_{j=1}^{i-1} \mathcal{G}^j),$$

where inequality $(a)$ holds by submodularity, and inequality $(b)$ holds by design of the greedy algorithm (Line 7 of Algorithm 3).

Combining Equation (2) and Equation (3) finishes the proof:

$$f(\mathcal{O}) \leq f(\mathcal{G}) + \sum_{t=1}^T f(\bigcup_{j=1}^i \mathcal{G}^j - \bigcup_{j=1}^{i-1} \mathcal{G}^j) = 2f(\mathcal{G}) - f(\mathcal{G}^0) \leq 2f(\mathcal{G}). \tag{4}$$

Furthermore, we prove that the competitive ratio of $1/2$ is tight for Problem 2.2.

Consider a simple adversarial example with budget $k = 1$ and a stream given by

$$\mathbf{S} = ((V_1, 0, 1), (V_2, 1, 1), (V_3, 1, 1)),$$

where $V_1 = \{c_1, c_2\}$ and $V_2 = \{c_3, c_4\}$. For any algorithm, if it selects $\mathcal{S}^1 = \{V_1\}$ as the first set to present, the adversary sets $V_3 = \{c_1, c_2\}$. Conversely, if the algorithm selects $\mathcal{S}^1 = \{V_2\}$, then the adversary sets $V_3 = \{c_3, c_4\}$.

In either case, the algorithm can only cover two topics, while the optimal solution covers all four. Thus, the competitive ratio is at most $\frac{2}{4} = \frac{1}{2}$, which completes the proof.

$\square$

## C   Omitted proofs of Section 3

Before presenting the omitted proofs in Section 3, we provide a detailed description of the reductions used in our analysis.

We begin by formally defining the streaming submodular maximization problem subject to a partition matroid constraint (*S2MM*). Our definition adapts the streaming submodular maximization framework under a $p$-matchoid constraint, as introduced by Chekuri et al. [8], and is stated as follows:

**Problem C.1** (*S2MM*). *Let $\mathcal{M} = (\bar{\mathbf{S}}, \mathcal{I})$ be a partition matroid and $\hat{f}$ a submodular function. The elements of $\bar{\mathbf{S}}$ are presented in a stream, and we order $\bar{\mathbf{S}}$ by order of appearance. The goal of the* S2MM *problem is to select a subset of items $\mathcal{A} \in \mathcal{I}$ that maximizes $\hat{f}(\mathcal{A})$.*

We demonstrate two types of reductions from an instance of *S3MOR* to an instance of *S2MM*, depending on whether we know the exact value of $T$. Algorithm 1, as presented in the main content, is based on the second reduction, where we only know an upper bound $T'$ of $T$.

### C.1   The case when the exact number of visits $T$ is known

First, we show that if the exact number of user visits $T$ is known, the *S3MOR* problem reduces to the *S2MM* problem. Specifically, any streaming algorithm for *S2MM* that satisfies the *irrevocable output requirement* can solve *S3MOR* while maintaining the same competitive ratio. The *irrevocable output requirement* requires that the output subset for any partition becomes fixed irrevocably once all its items have appeared in the stream.

**The reduction.**   Given an instance of the *S3MOR* problem, where the stream is given by

$$\mathbf{S} = ((V_1, p_1, \tau_1), \ldots, (V_N, p_N, \tau_N)),$$

with a budget $k$, and the function $f$ as defined in Equation (1), we can construct a corresponding instance of the *S2MM* problem.

Recall that $r_t$ denotes the time step at which the user submits the $t$-th visit (with $r_0 = 0$), and $V_i^t$ denotes a copy of item $V_i$ with copy identifier $t$. We use $\bigsqcup$ to denote the concatenation of streams. An instance of the *S2MM* problem with input stream $\bar{\mathbf{S}}$ can be constructed as follows:

1. Let $\bar{\mathbf{S}}_t = \bigsqcup_{i=r_{t-1}+1}^{r_t} (V_i^t, \cdots, V_i^T)$ for all $t \in [T]$, to be specific,

$$\bar{\mathbf{S}}_t = \left( \underbrace{V_{r_{t-1}+1}^t, \cdots, V_{r_{t-1}+1}^T}_{i=r_{t-1}+1}, \underbrace{V_{r_{t-1}+2}^t, \cdots, V_{r_{t-1}+2}^T}_{i=r_{t-1}+2}, \cdots, \underbrace{V_{r_t}^t, \cdots, V_{r_t}^T}_{i=r_t} \right).$$

2. Let $\bar{\mathbf{S}} = \bigsqcup_{t=1}^{T} \bar{\mathbf{S}}_t$ be the concatenation of sub-streams $\bar{\mathbf{S}}_1, \cdots, \bar{\mathbf{S}}_T$, i.e., $\bar{\mathbf{S}} = (\bar{\mathbf{S}}_1, \cdots, \bar{\mathbf{S}}_T)$.
3. Let $\mathcal{V}^t = \{V_i^t\}_{i=1}^{r_t} = \{V_1^t, \ldots, V_{r_t}^t\}$.
4. Construct $\mathcal{I}$ in the following way: for any set $X \subseteq \bigcup_{t=1}^{T} \mathcal{V}^t$, if $|X \cap \mathcal{V}^t| \le k$ for all $1 \le t \le T$, then $X \in \mathcal{I}$.

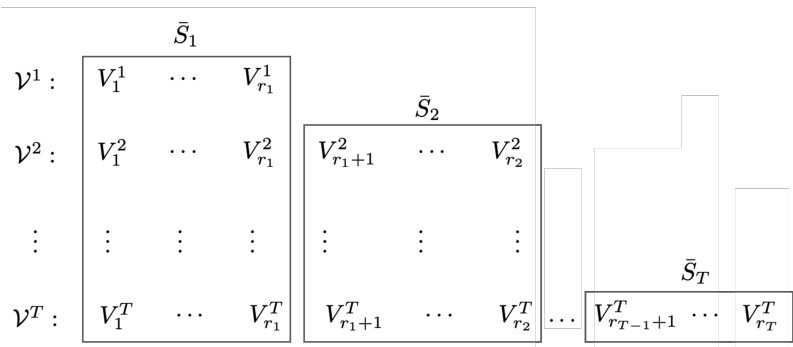

Figure 5: Illustration of the reduction when the exact number of visits $T$ is known. The original item stream for the *S3MOR* problem is $(V_1, \ldots, V_{r_T})$, with user accesses occurring at time steps $r_1, \ldots, r_T$. The constructed stream for the *S2MM* problem is $\bar{\mathbf{S}} = \bigsqcup_{t=1}^{T} \bar{\mathbf{S}}_t$.

5. Let $p_i^t = p_i$ denote a copy of $p_i$ with copy identifier $t \in [T]$. Define $\hat{f} : 2^{\bigcup_{t=1}^{T} \mathcal{V}^t} \to \mathbb{R}_{\geq 0}$. Specifically, for any $\mathcal{A} \subseteq \bigcup_{t=1}^{T} \mathcal{V}^t$, let $\mathcal{A}^t = \mathcal{A} \cap \mathcal{V}^t$. Then, $\hat{f}$ is defined as follows:

$$\hat{f}(\mathcal{A}) = \sum_{j=1}^{d} \left( 1 - \prod_{t \in [T]} \prod_{V_i^t \in \mathcal{A}^t, V_i^t \ni c_j} (1 - p_i^t) \right), \tag{5}$$

and the goal of *S2MM* is to find $\mathcal{A} \in \mathcal{I}$ that maximizes $\hat{f}(\mathcal{A})$.

We observe that $\mathcal{M} = (\bar{\mathbf{S}}, \mathcal{I})$, as defined above, forms a partition matroid on the constructed stream $\bar{\mathbf{S}}$, with $\bar{\mathbf{S}} = \bigcup_{t=1}^{T} \mathcal{V}^t$ and $\mathcal{V}^i \cap_{i \neq j} \mathcal{V}^j = \emptyset$. By construction, for each visit $t \in [T]$, the partition $\mathcal{V}^t$ contains exactly the set of items that arrived in the system before the $t$-th user visit in the *S3MOR* problem. Since the *S2MM* problem and the *S3MOR* problem share equivalent objective functions and equivalent constraints, we conclude the following: for any optimal solution $\bar{\mathcal{O}} \in \mathcal{I}$ of the *S2MM* problem, the sequence of subsets

$$\bar{\mathcal{O}} \cap \mathcal{V}^1, \ldots, \bar{\mathcal{O}} \cap \mathcal{V}^T$$

yields optimal output for the *S3MOR* problem at each corresponding $t$-th user visit.

We can further conclude that any streaming algorithm achieving an $\alpha$-competitive solution for the *S2MM* problem also yields an $\alpha$-competitive solution for the *S3MOR* problem. For this equivalence to hold, the streaming algorithm for *S2MM* must satisfy the *irrevocable output requirement*: for each partition $\mathcal{V}^t$ of the stream, once the final item from this partition has arrived, the solution that belongs to this partition, $\mathcal{A} \cap \mathcal{V}^t$, can no longer be changed. This requirement excludes algorithms like that of Feldman et al. [11], whose solution set only becomes available after the stream terminates. In contrast, the algorithms proposed by Chakrabarti and Kale [7], Chekuri et al. [8], and Feldman et al. [10] are suitable. These three algorithms all meet the requirement while providing $1/4$-competitive solutions for the *S2MM* problem.

## C.2 The case when only an upper bound $T'$ of $T$ is available

Consider the case where only an upper bound $T'$ of the true number of user visits $T$ is known. We prove the following: (1) An instance of the *S2MM* problem can be constructed such that any feasible solution to the *S2MM* problem contains a valid solution for the *S3MOR* problem. (2) Any streaming algorithm that solves the *S2MM* problem and satisfies the *irrevocable output requirement* can be adapted to solve the *S3MOR* problem, albeit with a reduced competitive ratio. Specifically, the competitive ratio decays by a factor of $1/(T' - T + 1)$.

In contrast to the reduction we present in Appendix C.1, the construction of the stream for the *S2MM* problem depends on both the original stream $\mathbf{S}$, and the algorithmic procedure described in Algorithm 1. We will prove that:

1. Algorithm 1 generates a corresponding stream $\hat{\mathbf{S}}$ for the *S2MM* problem

| | $\hat{\mathbf{S}}_1$ | | $\hat{\mathbf{S}}_2$ | | $\hat{\mathbf{S}}_3$ | |
|---|---|---|---|---|---|---|
| $\mathcal{G}^2 \subseteq \hat{\mathcal{V}}^1:$ | $V_1^1$ $\cdots$ | $V_{r_1}^1$ | $V_{r_1+1}^1$ $\cdots$ | $V_{r_2}$ | | |
| $\mathcal{G}^3 \subseteq \hat{\mathcal{V}}^2:$ | $V_1^2$ $\cdots$ | $V_{r_1}^2$ | $V_{r_1+1}^2$ $\cdots$ | $V_{r_2}^2$ | $V_{r_2+1}^2$ $\cdots$ | $V_{r_3}^2$ |
| $\hat{\mathcal{V}}^3:$ | $V_1^3$ $\cdots$ | $V_{r_1}^3$ | $V_{r_1+1}^3$ $\cdots$ | $V_{r_2}^3$ | $V_{r_2+1}^3$ $\cdots$ | $V_{r_3}^3$ |
| $\mathcal{G}^1 \subseteq \hat{\mathcal{V}}^4:$ | $V_1^4$ $\cdots$ | $V_{r_1}^4$ | | | | |
| $\hat{\mathcal{V}}^5:$ | $V_1^5$ $\cdots$ | $V_{r_1}^5$ | $V_{r_1+1}^5$ $\cdots$ | $V_{r_2}^5$ | $V_{r_2+1}^5$ $\cdots$ | $V_{r_3}^5$ |

Figure 6: An example illustrating the reduction when only an upper bound $T'$ of the true number of visits $T$ is available. In this example, $T = 3$ and $T' = 5$. The original item stream for the *S3MOR* problem is given by $(V_1, V_2, \ldots, V_{r_3})$, with user visits occurring at time steps $r_1$, $r_2$, and $r_3$. The constructed stream for the *S2MM* problem is $\hat{\mathbf{S}} = \bigsqcup_{t=1}^3 \hat{\mathbf{S}}_t$, where $\hat{\mathbf{S}}_1 = \bigsqcup_{i=1}^{r_1} \bigsqcup_{a \in [5]} (V_i^a)$, $\hat{\mathbf{S}}_2 = \bigsqcup_{i=r_1+1}^{r_2} \bigsqcup_{a \in \{1,2,3,5\}} (V_i^a)$, and $\hat{\mathbf{S}}_3 = \bigsqcup_{i=r_2+1}^{r_3} \bigsqcup_{a \in \{2,3,5\}} (V_i^a)$. Algorithm 1 maintains and updates $\mathcal{A}^t \in \hat{\mathcal{V}}^t$ for all $t \in [5]$ upon the arrival of each item in $\hat{\mathbf{S}}$. At the visit time steps $r_1$, $r_2$, and $r_3$, it outputs $\mathcal{G}^1 = \mathcal{A}^4$, $\mathcal{G}^2 = \mathcal{A}^1$, and $\mathcal{G}^3 = \mathcal{A}^2$, respectively. By the end of the stream, the union $\bigcup_{t=1}^5 \mathcal{A}^t$ constitutes a $1/4$-competitive solution for the *S2MM* problem on input $\hat{\mathbf{S}}$. The combined solution $\bigcup_{t=1}^3 \mathcal{G}^t$ is a $\frac{1}{4 \times (5-3+1)}$-competitive solution for the original *S3MOR* problem.

2. The candidate sets $\{\mathcal{A}^1, \cdots, \mathcal{A}^{T'}\}$ maintained by Algorithm 1 forms a solution to the *S2MM* problem; the output $\{\mathcal{G}^1, \cdots, \mathcal{G}^T\}$ of Algorithm 1 forms a solution to the *S3MOR* problem.

By Algorithm 1 line 13, we can construct a bijective mapping $\pi$ between the index sets of $\{\mathcal{G}^1, \ldots, \mathcal{G}^T\}$ and $\{\mathcal{A}^1, \ldots, \mathcal{A}^{T'}\}$. This mapping satisfies the equality:

$$\mathcal{G}^t = \mathcal{A}^{\pi(t)}, \quad \text{for all } t \in [T].$$

Let $\mathcal{T}^0 = \{1, 2, \ldots, T'\}$ denote the initial index set of all active candidate sets. For each $t \geq 1$, let $\mathcal{T}^t$ represent the index set of active candidate sets before the $t$-th visit and after the $(t-1)$-th visit. The active index sets evolve as follows:

(1) For all $t \in [T]$: $\mathcal{T}^t = \mathcal{T}^{t-1} \setminus \{\pi(t)\}$, and

(2) for all $t \in \{T+1, \ldots, T'\}$: $\mathcal{T}^t = \mathcal{T}^T$.

**The reduction.**

1. We construct the stream $\hat{\mathbf{S}}$ for *S2MM* as follows

$$\hat{\mathbf{S}}_t = \begin{cases} \bigsqcup_{i=r_{t-1}+1}^{r_t} \bigsqcup_{a \in \mathcal{T}^t} (V_i^a), & \text{for } t \in [T], \\ \bigsqcup_{i=r_{T-1}+1}^{r_T} \bigsqcup_{a \in \mathcal{T}^T} (V_i^a), & \text{for } t \in \{T+1, \cdots, T'\}. \end{cases}$$

2. Let $\hat{\mathbf{S}} = \bigsqcup_{t=1}^T \hat{\mathbf{S}}_t$ be the concatnation of $\hat{\mathbf{S}}_1, \cdots, \hat{\mathbf{S}}_T$.

3. Let $\rho(t) = \pi^{-1}(t)$, and

$$\hat{\mathcal{V}}^t = \begin{cases} \{V_i^t\}_{i=1}^{r_{\rho(t)}} = \{V_1^t, \ldots, V_{r_{\rho(t)}}^t\}, & \text{for } t \in \{\pi(1), \cdots, \pi(T)\}, \\ \{V_1^t, \cdots, V_{r_T}^t\}, & \text{for } t \in \mathcal{T}^0 \setminus \{\pi(1), \cdots, \pi(T)\}. \end{cases}$$

4. Construct $\hat{\mathcal{I}}$ in the following way: for any set $X \subseteq \bigcup_{t=1}^{T'} \hat{\mathcal{V}}^t$, if $|X \cap \hat{\mathcal{V}}^t| \leq k$ for all $1 \leq t \leq T'$, then $X \in \hat{\mathcal{I}}$.

5. Let $p_i^t = p_i$ denote a copy of $p_i$ with copy identifier $t \in [T']$. The goal is to select $\mathcal{A} = \bigcup_{t=1}^{T'} \mathcal{A}^t \in \hat{\mathcal{I}}$ that maximizes $f(\mathcal{A})$. The submodular function $\hat{f}$ is given as follows:

$$\hat{f}(\mathcal{A}) = \sum_{j=1}^{d} \left( 1 - \prod_{t \in [T']} \prod_{V_i^t \in \mathcal{A}^t, V_i^t \ni c_j} (1 - p_i^t) \right). \tag{6}$$

Given the above construction, we can verify that $\mathcal{M} = (\hat{\mathbf{S}}, \hat{\mathcal{I}})$ is an instance of a partition matroid defined on the constructed stream $\hat{\mathbf{S}}$, with $\hat{\mathbf{S}} = \bigcup_{t=1}^{T'} \hat{\mathcal{V}}^t$ and $\hat{\mathcal{V}}^i \cap_{i \neq j} \hat{\mathcal{V}}^j = \emptyset$. By construction, the partitions $\hat{\mathcal{V}}^{\pi(t)}$ of $\hat{\mathbf{S}}$ satisfy: (1) for each $t \in [T]$, $\hat{\mathcal{V}}^{\pi(t)}$ contains exactly the set of items that arrived before the $t$-th user visit in the *S3MOR* problem (2) The remaining $T' - T$ partitions each contain the same set of items that arrived before the final user visit. The design ensures that each output $\mathcal{G}^t$ satisfies $\mathcal{G}^t \subseteq \hat{\mathcal{V}}^{\pi(t)}$ and $|\mathcal{G}^t| \leq k$, for all $t \in [T]$. Consequently, the sequence $\mathcal{G}^1, \dots, \mathcal{G}^T$ constitutes feasible output for the *S3MOR* problem.

Following Algorithm 1, we conclude that for all $t \in [T']$, we have $\mathcal{A}^t \subseteq \hat{\mathcal{V}}^t$ with $|\mathcal{A}^t| \leq k$. Consequently, the union $\bigcup_{t=1}^{T'} \mathcal{A}^t$ forms a feasible solution to the *S2MM* problem.

Since Algorithm 1 essentially applies the streaming algorithm of Chekuri et al. [8] to the constructed stream $\hat{\mathbf{S}}$, the solution $\bigcup_{t=1}^{T'} \mathcal{A}^t$ achieves a $1/4$-approximation guarantee for the *S2MM* problem.

### C.3 Omitted proofs

**Theorem 3.1.** *Let $T$ be the number of user accesses and $T'$ a given upper bound. Algorithm 1 has competitive ratio $\frac{1}{4(T'-T+1)}$, space complexity $\mathcal{O}(T'k)$, time complexity $\mathcal{O}(NkT')$, and response time complexity $\mathcal{O}(T')$.*

*Proof.* Following the notations and conclusions we have in Appendix C.2, we are ready to prove Theorem 3.1. Let $\mathcal{O}^1, \cdots, \mathcal{O}^T$ be the optimal solution for the *S3MOR* problem, and let $\hat{\mathcal{O}} = \bigcup_{t=1}^{T'} \hat{\mathcal{O}}^t$, where $\hat{\mathcal{O}}^t = \hat{\mathcal{O}} \cap \hat{\mathcal{V}}^t$, denotes the optimal solution for the *S2MM*. We have

$$f(\bigcup_{t=1}^{T} \mathcal{G}^t) = f(\bigcup_{t=1}^{T-1} \mathcal{A}^{\pi(t)} + \mathcal{A}^{\pi(T)}) = f(\mathcal{A}^{\pi(T)} \mid \bigcup_{t=1}^{T-1} \mathcal{A}^{\pi(t)}) + f(\bigcup_{t=1}^{T-1} \mathcal{A}^{\pi(t)}) \tag{7}$$

$$\overset{(a)}{\geq} \frac{1}{T'-T+1} \sum_{s \in \mathcal{T} \setminus \cup_{t=1}^{T-1} \{\pi(t)\}} f(\mathcal{A}^s \mid \bigcup_{t=1}^{T-1} \mathcal{A}^{\pi(t)}) + f(\bigcup_{t=1}^{T-1} \mathcal{A}^{\pi(t)})$$

$$\geq \frac{1}{T'-T+1} \sum_{s \in \mathcal{T} \setminus \cup_{t=1}^{T-1} \{\pi(t)\}} f(\mathcal{A}^s \mid \bigcup_{t=1}^{T-1} \mathcal{A}^{\pi(t)}) + \frac{1}{T'-T+1} f(\bigcup_{t=1}^{T-1} \mathcal{A}^{\pi(t)})$$

$$\overset{(b)}{\geq} \frac{1}{T'-T+1} f(\bigcup_{t=1}^{T'} \mathcal{A}^t) \overset{(c)}{\geq} \frac{1}{4(T'-T+1)} f(\bigcup_{t=1}^{T'} \hat{\mathcal{O}}^t) \overset{(d)}{\geq} \frac{1}{4(T'-T+1)} f(\bigcup_{t=1}^{T} \mathcal{O}^t),$$

where inequality $(a)$ holds because by line 11 of Algorithm 1, that $\mathcal{A}^{\pi(T)}$ is selected with the largest marginal gain in terms of $f(\bigcup_{t=1}^{T-1} \mathcal{A}^{\pi(t)})$. Inequality $(b)$ holds by submodularity. Inequality $(c)$ holds because, as concluded in Appendix C.2, $\bigcup_{t=1}^{T'} \mathcal{A}^t$ achieves a $1/4$-approximation guarantee for the *S2MM* problem. Finally, inequality $(d)$ holds by contradiction: if $f(\bigcup_{t=1}^{T} \mathcal{O}^t) > f(\bigcup_{t=1}^{T'} \hat{\mathcal{O}}^t)$, then there exists a feasible solution for the *S2MM* problem where for all $t \in [T]$, $\mathcal{A}^t = \mathcal{O}^t$, and regardless of the choice of $\mathcal{A}^{T+1}, \cdots, \mathcal{A}^T$, it holds that $f(\bigcup_{t=1}^{T'} \mathcal{A}^t) \geq f(\bigcup_{t=1}^{T} \mathcal{O}^t) > f(\bigcup_{t=1}^{T'} \hat{\mathcal{O}}^t)$, which contradicts that $\hat{\mathcal{O}}$ is an optimal solution.

**Complexity analysis.** The space complexity is in $\mathcal{O}(T'k)$ because STORM only needs to maintain $T'$ candidate sets of size $k$. Upon the arrival of each item, STORM makes $T'k$ oracle calls, thus the

time complexity is in $\mathcal{O}(NT'k)$. When a user submit a visit request, STORM selects one partition out of at most $T'$ partitions and output it, thus the response time complexity is in $\mathcal{O}(T')$.

$\square$

**Tightness of our results.** Note that the analysis in Equation (7) is tight. In the following, we show that for any value of $T$, we can construct a stream such that STORM cannot do better than $\frac{1}{T'-T+1}$, which is only a constant factor away from our analysis in Theorem 3.1.

To see this, consider an instance of the *S3MOR* problem with budget $k = 1$, the user visits one time at the end of the stream, and we have an estimated number of user visits $T' > 1$, the stream is constructed as below:

$$\mathbf{S}_1 = ((\{1\}, 1, 0), (\{2\}, 1, 0), \cdots, (\{T'\}, 1, 0), (\{1, \cdots, T'\}, 1, 1))$$

The STORM($T'$) algorithm outputs one of the first $T'$ items and achieves an expected topic coverage of value 1, while the optimal solution is to select the last item and obtain an expected coverage of value $T'$. The competitive ratio in this instance is $\frac{1}{T'}$, while Alg. 1 has competitive ratio $\frac{1}{4(T'-T+1)} = \frac{1}{4T'}$.

Note that for all possible numbers of user visits $T$ and $T' > T$, we can construct a stream such that STORM cannot do better than $\frac{1}{T'-T+1}$. For example, when $T = 2$, we can construct a sub-stream $\mathbf{S}_2$ as follows

$$\mathbf{S}_2 = ((\{T'+1, T'+2\}, 1, 0), \cdots, (\{2T'-1, 2T'\}, 1, 0), (\{T'+1, \cdots, 3T'-3\}, 1, 1))$$

By concatenating $\mathbf{S}_1$ and $\mathbf{S}_2$, we obtain a new stream where the user visits for 2 times. With budget $k = 1$, STORM($T'$) achieves an expected coverage 3, while the optimal expected coverage is $3(T'-1)$, leading to a competitive ratio of $\frac{1}{T'-1}$, with is again a 1/4 factor away from $\frac{1}{4(T'-T+1)} = \frac{1}{T'-1}$. We can repeat this process such that for any $T$, we can construct a stream such that the competitive ratio is equal to $\frac{1}{T'-T+1}$.

**Lemma C.2.** *Let $T$ be the number of user accesses and $T'$ be a given upper bound. Algorithm 2 generates a set of guesses of $T$ as $\mathcal{P} = \{\delta i \mid i \in [\lceil T'/\delta \rceil]\}$. Let $T^*$ be the smallest integer in $\mathcal{P}$ that is larger than or equal to $T$. Let $\mathcal{G}^1, \cdots, \mathcal{G}^T$ be the output of STORM($T^*$), and let $\mathcal{B}^1, \cdots, \mathcal{B}^T$ be the output of Algorithm 2, then it holds that*

$$f(\bigcup_{t=1}^{T} \mathcal{B}^t) \geq f(\bigcup_{t=1}^{T} \mathcal{G}^t) + \sum_{t=1}^{T-1} \left[ f(\bigcup_{s=1}^{t} \mathcal{B}^s \mid \bigcup_{s=1}^{t+1} \mathcal{G}^s) - f(\bigcup_{s=1}^{t} \mathcal{B}^s \mid \bigcup_{s=1}^{t} \mathcal{G}^s) \right]. \tag{8}$$

*Proof.* We prove Equation (8) via induction.

When $T = 1$, according to line 7 of Algorithm 2 that $\mathcal{B}^1$ is chosen with the largest marginal gain, it holds that $f(\mathcal{B}^1) \geq f(\mathcal{G}^1)$.

The base case for the induction starts from $T = 2$. It is

$$
\begin{aligned}
f(\mathcal{B}^1, \mathcal{B}^2) = f(\mathcal{B}^1) + f(\mathcal{B}^2 \mid \mathcal{B}^1) &\overset{(a)}{\geq} f(\mathcal{G}^1) + f(\mathcal{G}^2 \mid \mathcal{B}^1) \\
&= f(\mathcal{G}^1 \cup \mathcal{B}^1) - f(\mathcal{B}^1 \mid \mathcal{G}^1) + f(\mathcal{G}^2 \mid \mathcal{B}^1) \\
&= f(\mathcal{B}^1) + f(\mathcal{G}^1 \mid \mathcal{B}^1) - f(\mathcal{B}^1 \mid \mathcal{G}^1) + f(\mathcal{G}^2 \mid \mathcal{B}^1) \\
&\overset{(b)}{\geq} f(\mathcal{G}^1 \cup \mathcal{G}^2 \mid \mathcal{B}^1) + f(\mathcal{B}^1) - f(\mathcal{B}^1 \mid \mathcal{G}^1) = f(\mathcal{B}^1, \mathcal{G}^1, \mathcal{G}^2) - f(\mathcal{B}^1 \mid \mathcal{G}^1) \\
&= f(\mathcal{G}^1, \mathcal{G}^2) + f(\mathcal{B}^1 \mid \mathcal{G}^1 \cup \mathcal{G}^2) - f(\mathcal{B}^1 \mid \mathcal{G}^1),
\end{aligned}
$$

where inequality (a) holds because $f(\mathcal{B}^1) \geq f(\mathcal{G}^1)$ and $f(\mathcal{B}^2 \mid \mathcal{B}^1) > f(\mathcal{G}^2 \mid \mathcal{B}^1)$. Inequality (b) holds by submodularity.

We then prove the induction step. Assume for $T = j$, it holds

$$f(\bigcup_{t=1}^{j} \mathcal{B}^t) \geq f(\bigcup_{t=1}^{j} \mathcal{G}^t) + \sum_{t=1}^{j-1} \left[ f(\bigcup_{s=1}^{t} \mathcal{B}^s \mid \bigcup_{s=1}^{t+1} \mathcal{G}^s) - f(\bigcup_{s=1}^{t} \mathcal{B}^s \mid \bigcup_{s=1}^{t} \mathcal{G}^s) \right]. \tag{9}$$

We can use $\phi_j$ to represent the second term in Equation (9), i.e., $f(\bigcup_{t=1}^{j} \mathcal{B}^t) \geq f(\bigcup_{t=1}^{j} \mathcal{G}^t) + \phi_j$, we can then show that for $T = j+1$ it holds

$$
\begin{aligned}
f(\bigcup_{t=1}^{j+1} \mathcal{B}^t) &= f(\mathcal{B}^{j+1} \mid \bigcup_{t=1}^{j} \mathcal{B}^t) + f(\bigcup_{t=1}^{j} \mathcal{B}^t) \overset{(a)}{\geq} f(\mathcal{G}^{j+1} \mid \bigcup_{t=1}^{j} \mathcal{B}^t) + f(\bigcup_{t=1}^{j} \mathcal{G}^t) + \phi_j \\
&= f(\bigcup_{t=1}^{j} \mathcal{G}^j \mid \bigcup_{t=1}^{j} \mathcal{B}^t) + f(\bigcup_{t=1}^{j} \mathcal{B}^t) - f(\bigcup_{t=1}^{j} \mathcal{B}^t \mid \bigcup_{t=1}^{j} \mathcal{G}^t) + f(\mathcal{G}^{j+1} \mid \bigcup_{t=1}^{j} \mathcal{B}^t) + \phi_j \\
&\overset{(b)}{\geq} f(\bigcup_{t=1}^{j+1} \mathcal{G}^t \mid \bigcup_{t=1}^{j} \mathcal{B}^t) + f(\bigcup_{t=1}^{j} \mathcal{B}^t) - f(\bigcup_{t=1}^{j} \mathcal{B}^t \mid \bigcup_{t=1}^{j} \mathcal{G}^t) + \phi_j \\
&= f(\bigcup_{t=1}^{j+1} \mathcal{G}^t) + f(\bigcup_{t=1}^{j} \mathcal{B}^t \mid \bigcup_{t=1}^{j+1} \mathcal{G}^t) - f(\bigcup_{t=1}^{j} \mathcal{B}^t \mid \bigcup_{t=1}^{j} \mathcal{G}^t) + \phi_j \\
&= f(\bigcup_{t=1}^{j+1} \mathcal{G}^j) + \sum_{t=1}^{j} \left[ f(\bigcup_{s=1}^{t} \mathcal{B}^s \mid \bigcup_{s=1}^{t+1} \mathcal{G}^s) - f(\bigcup_{s=1}^{t} \mathcal{B}^s \mid \bigcup_{s=1}^{t} \mathcal{G}^s) \right].
\end{aligned}
$$

Here, inequality $(a)$ holds by the induction assumption, and $(b)$ holds by submodularity. This completes the induction. $\qquad\square$

**Lemma C.3.** *Let $T$ be the number of user accesses and $T'$ be a given upper bound. Algorithm 2 generates a set of guesses of $T$ as $\mathcal{P} = \{\delta i \mid i \in [\lceil T'/\delta \rceil]\}$. Let $T^*$ be the smallest integer in $\mathcal{P}$ that is larger than or equal to $T$. Let $\mathcal{G}^1, \cdots, \mathcal{G}^T$ be the output of $\mathrm{STORM}(T^*)$, and let $\mathcal{B}^1, \cdots, \mathcal{B}^T$ be the output of Algorithm 2. It holds that*

$$
f(\bigcup_{t=1}^{T} \mathcal{B}^t) \geq \frac{1}{2} f(\bigcup_{t=1}^{T} \mathcal{G}^t). \tag{10}
$$

*Proof.* Lemma C.3 is a direct result of Lemma C.2, as we can show that

$$
\sum_{t=1}^{T-1} \left[ f(\bigcup_{s=1}^{t} \mathcal{B}^s \mid \bigcup_{s=1}^{t+1} \mathcal{G}^s) - f(\bigcup_{s=1}^{t} \mathcal{B}^s \mid \bigcup_{s=1}^{t} \mathcal{G}^s) \right] \geq -f(\bigcup_{t=1}^{T} \mathcal{B}^t).
$$

To prove that the above inequality holds, we proceed to show that

$$
\sum_{t=1}^{T-1} \left[ -f(\bigcup_{s=1}^{t} \mathcal{B}^s \mid \bigcup_{s=1}^{t+1} \mathcal{G}^s) + f(\bigcup_{s=1}^{t} \mathcal{B}^s \mid \bigcup_{s=1}^{t} \mathcal{G}^s) \right] \leq f(\bigcup_{t=1}^{T} \mathcal{B}^t).
$$

Indeed, it is

$$\sum_{t=1}^{T-1}\Big[-f\big(\bigcup_{s=1}^{t}\mathcal{B}^s \mid \bigcup_{s=1}^{t+1}\mathcal{G}^s\big)+f\big(\bigcup_{s=1}^{t}\mathcal{B}^s \mid \bigcup_{s=1}^{t}\mathcal{G}^s\big)\Big] \tag{11}$$

$$=\sum_{t=1}^{T-1}\Big[f\big(\bigcup_{s=1}^{t}\mathcal{B}^s \mid \bigcup_{s=1}^{t}\mathcal{G}^s\big)-f\big(\bigcup_{s=1}^{t}\mathcal{B}^s \mid \bigcup_{s=1}^{t+1}\mathcal{G}^s\big)\Big]$$

$$=f(\mathcal{B}^1 \mid \mathcal{G}^1)+\sum_{t=1}^{T-2}\Big[f\big(\bigcup_{s=1}^{t+1}\mathcal{B}^s \mid \bigcup_{s=1}^{t+1}\mathcal{G}^s\big)-f\big(\bigcup_{s=1}^{t}\mathcal{B}^s \mid \bigcup_{s=1}^{t+1}\mathcal{G}^s\big)\Big]-f\big(\bigcup_{s=1}^{T-1}\mathcal{B}^s \mid \bigcup_{s=1}^{T}\mathcal{G}^s\big)$$

$$\le f(\mathcal{B}^1)+\sum_{t=1}^{T-2}\Big[f\big(\bigcup_{s=1}^{t+1}\mathcal{B}^s \mid \bigcup_{s=1}^{t+1}\mathcal{G}^s\big)-f\big(\bigcup_{s=1}^{t}\mathcal{B}^s \mid \bigcup_{s=1}^{t+1}\mathcal{G}^s\big)\Big]$$

$$=f(\mathcal{B}^1)+\sum_{t=1}^{T-2}\Big[f\big(\bigcup_{s=1}^{t+1}\mathcal{B}^s + \bigcup_{s=1}^{t+1}\mathcal{G}^s\big)-f\big(\bigcup_{s=1}^{t+1}\mathcal{G}^s\big)-f\big(\bigcup_{s=1}^{t}\mathcal{B}^s + \bigcup_{s=1}^{t+1}\mathcal{G}^s\big)+f\big(\bigcup_{s=1}^{t+1}\mathcal{G}^s\big)\Big]$$

$$=f(\mathcal{B}^1)+\sum_{t=1}^{T-2}f\big(\mathcal{B}^{t+1} \mid \bigcup_{s=1}^{t}\mathcal{B}^s + \bigcup_{s=1}^{t+1}\mathcal{G}^s\big)\le f(\mathcal{B}^1)+\sum_{t=1}^{T-2}f\big(\mathcal{B}^{t+1} \mid \bigcup_{s=1}^{t}\mathcal{B}^s\big)$$

$$=f\big(\bigcup_{t=1}^{T-1}\mathcal{B}^t\big)\le f\big(\bigcup_{t=1}^{T}\mathcal{B}^t\big) \tag{12}$$

Combining Equation (12) with Equation (8), we can obtain Equation (10).

$\square$

**Theorem 3.2.** *Algorithm 2 has a competitive ratio of $1/(8\delta)$, space complexity $\mathcal{O}(T'^2 k/\delta)$, time complexity $\mathcal{O}(NkT'^2/\delta)$, and response time complexity $\mathcal{O}(T')$.*

*Proof.* Let $T^*$ be the smallest integer in $\mathcal{P}=\{\delta i \mid i \in [\lceil T'/\delta\rceil]\}$ that is larger than or equal to $T$. Let $\mathcal{G}^1,\cdots,\mathcal{G}^T$ be the output of STORM$(T^*)$

Observe that $T^* \le T+\delta-1$, otherwise we can set $T^*$ as $T^*-\delta$ and it is still an upper bound of $T$. By Theorem 3.1, it holds that

$$f\big(\bigcup_{t=1}^{T}\mathcal{G}^t\big)\ge \frac{1}{4(T^*-T+1)}f\big(\bigcup_{t=1}^{T}\mathcal{O}^t\big)\ge \frac{1}{4\delta}f\big(\bigcup_{t=1}^{T}\mathcal{O}^t\big). \tag{13}$$

Combining Equation (13) with Lemma C.3, we conclude that

$$f\big(\bigcup_{t=1}^{T}\mathcal{B}^t\big)\ge \frac{1}{2}f\big(\bigcup_{t=1}^{T}\mathcal{G}^t\big)\ge \frac{1}{8\delta}f\big(\bigcup_{t=1}^{T}\mathcal{O}^t\big). \tag{14}$$

Since STORM++ maintains $\lceil T'/\delta\rceil$ copies of STORM, each copy of STORM has time complexity in $\mathcal{O}(NkT')$ and space complexity in $\mathcal{O}(T'k)$, the space complexity for the STORM++ is $\mathcal{O}(T'^2 k/\delta)$, and the time complexity is $\mathcal{O}(NkT'^2/\delta)$. When a user submits a visit request, each copy of STORM selects one partition out of at most $T'$ partitions as a candidate result set, then STORM++ selects the best set among $T'/\delta$ candidates as output. Thus, the response time of STORM++ is $\mathcal{O}(T')$. $\square$

## D   Additional content for Section 4

### D.1   Dataset and experimental setting

We use the following six datasets:

- **KuaiRec** [14]: Recommendation logs from a video-sharing mobile app, containing metadata such as each video's categories and watch ratio. We use the *watch ratio*, computed as the user's viewing time divided by the video's duration, as the user rating score. The dataset includes 7 043 videos across 100 categories.
- **Anime:**[3] A dataset of animation ratings in the range $[1, 10]$, consisting of 7745 animation items across 43 genres.
- **Beer:**[4] This dataset contains BeerAdvocate reviews [32, 31], along with categorical attributes for each beer. After filtering out beers and reviewers with fewer than 10 reviews, the final dataset includes 9 000 beers across 70 categories.
- **Yahoo:**[5] A music rating dataset with ratings in the range $[1, 5]$. After filtering out users with fewer than 20 ratings, the dataset includes 136 736 songs across 58 genres.
- **RCV1** [28]: A multi-label dataset, each item is a news story from Reuters, Ltd. for research use. We sample 462 225 stories across 476 unique labels.
- **Amazon** [30]: A multi-label dataset consisting of 1 117 006 Amazon products over 3 750 unique labels. This dataset is a subset of the publicly available dataset.

**Experimental setting**    To obtain the click probabilities for each item, we generate them uniformly at random in the range $[0, 0.2]$ for the RCV1 and Amazon datasets. For the four item-rating datasets, we estimate click probabilities by computing a low-rank completion of the user-item rating matrix using matrix factorization [25]. This yields latent feature vectors $w_u$ for each user $u$ and $v_m$ for each item $m$, where the inner product $w_u^\top v_m$ approximates user $u$'s rating for item $m$. We then apply standard min-max normalization to the predicted ratings and linearly scale the results to the range $[0, 0.5]$ to obtain click probabilities. We set the latent feature dimension to 15 for the KuaiRec dataset, and to 20 for the remaining three rating datasets. While click probability ranges can vary in real-world scenarios, where larger ranges can lead to faster convergence toward the maximum expected coverage value, we intentionally set the probabilities to be small in our experiments. This allows us to observe performance changes over a wider range of parameter settings.

**Implementation**    To enhance the computational efficiency of the LMGREEDY algorithm (described in Algorithm 3), we implement lines 6–9 using the STOCHASTIC-GREEDY approach introduced by Mirzasoleiman et al. [35]. For the implementations of STORM and STORM++, we incorporate the sampling technique from Feldman et al. [10]. Specifically, when adding each item (with potential replacement) to each candidate set, we ignore it with probability 2/3.

### D.2    Omitted results

In this section, we present additional experimental results and analyses. Specifically, we show how the expected coverage changes as the budget $k$ varies in Figure 7, and as the number of user visits $T$ varies in Figure 8. The standard deviation of the expected coverage, as $k$ and $T$ are varied, is reported in Figure 10 and Figure 11, respectively. Finally, we present the effect of varying $\delta$ and $\Delta T$ on the expected coverage in Figure 9.

**Why STORM++ and STORM($T'$) outperform STORM($T$)?**    We present a simple example where STORM($T'$) outperforms STORM($T$). Consider the input stream $\mathbf{S} = ((\{1\}, 1, 0), (\{2\}, 1, 0), (\{3, 4\}, 0.9, 1), (\{5, 6\}, 1, 1))$ with budget $k = 1$ and actual number of visits $T = 2$. The algorithm STORM(2) selects either $\{1\}$ or $\{2\}$ for the first visit, and $\{5, 6\}$ for the second, achieving expected coverage of 3.

If we overestimate the visits and set $T' = 3$, STORM(3) instead selects $\{3, 4\}$ first and $\{5, 6\}$ second, yielding an expected coverage of 3.8, which is higher than when the number of visits is known exactly. Likewise, STORM++ with $T' = 3$ and $\delta = 3$ matches STORM(3), and achieves the same coverage of 3.8.

This example shows that while STORM($T$) has the strongest theoretical guarantee, STORM($T'$) and STORM++ can outperform it empirically, which explains the trends observed in our experiments.

---

[3]https://www.kaggle.com/datasets/CooperUnion/anime-recommendations-database
[4]https://cseweb.ucsd.edu/~jmcauley/datasets.html
[5]https://webscope.sandbox.yahoo.com/catalog.php?datatype=i&did=67

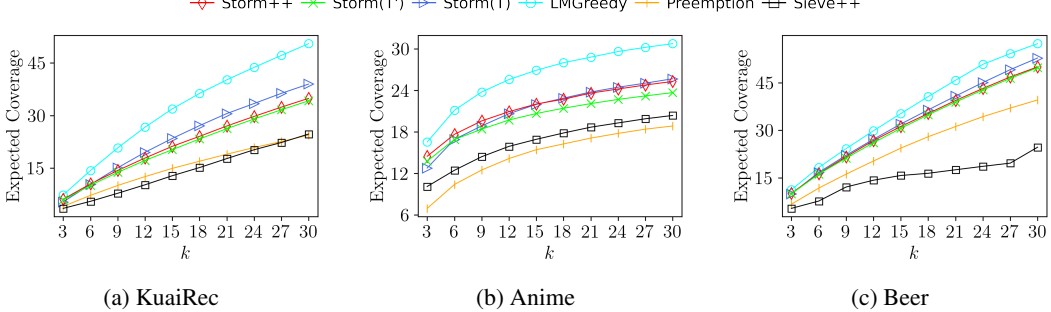

(a) KuaiRec        (b) Anime        (c) Beer

Figure 7: Empirical variation in expected coverage as a function of budget $k$ on three smaller datasets. Parameter $T = 5$, $\delta = 25$ and $\Delta T = 45$ are fixed

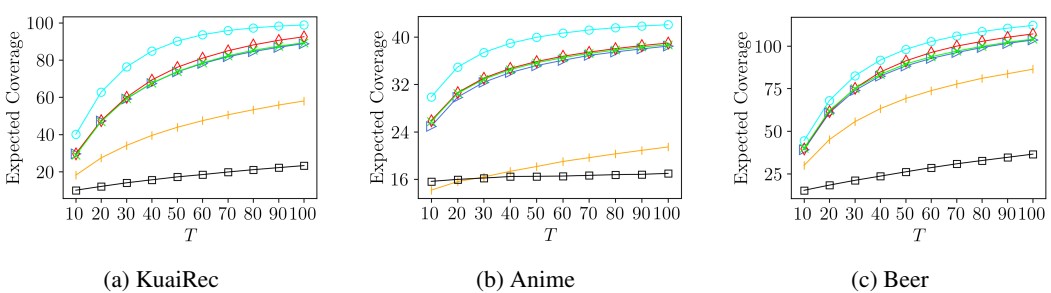

(a) KuaiRec        (b) Anime        (c) Beer

Figure 8: Empirical variation in expected coverage as a function of number of visits $T$ on three smaller datasets. Parameter $k = 10$, $\delta = 10$ and $\Delta T = 10$ are fixed.

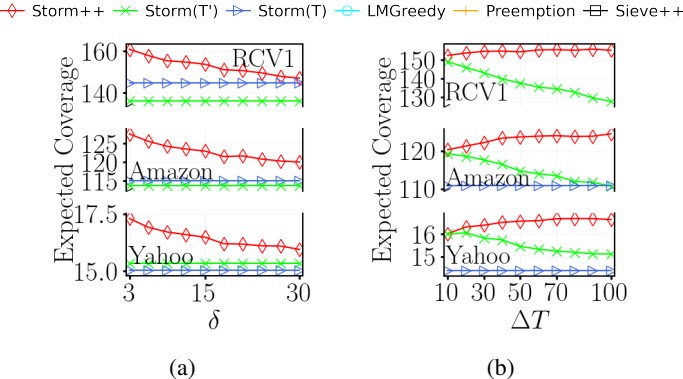

(a)           (b)

Figure 9: Impact of $\Delta T$ and $\delta$ on expected coverage on three larger datasets. In (a) and (b), we fix $k = 5$ and $T = 10$. When varying $\delta$, we fix $\Delta T = 60$; when varying $\Delta T$, we fix $\delta = 10$.

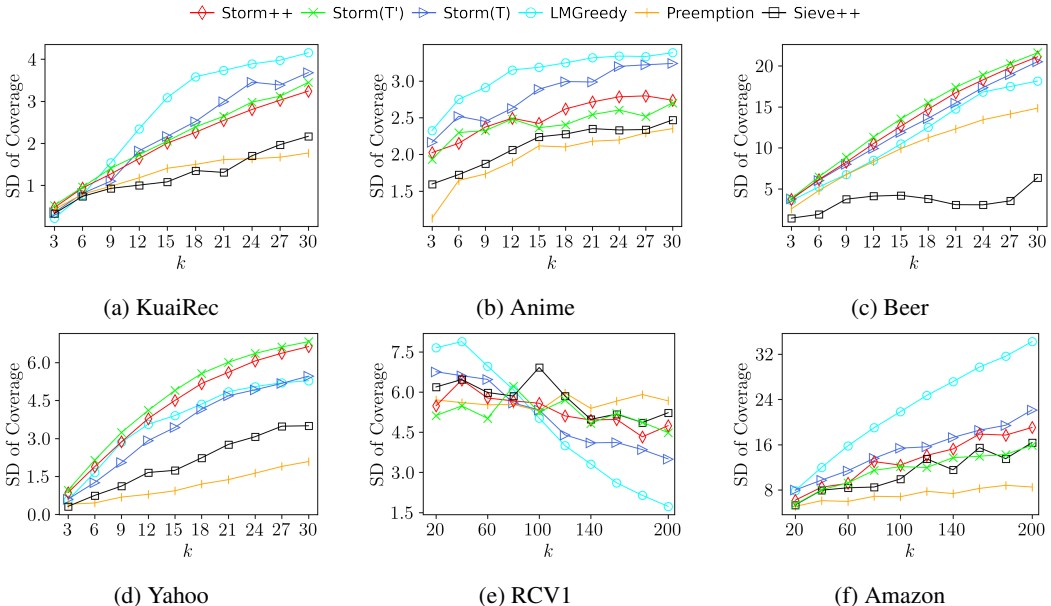

Figure 10: Empirical variation in standard deviation for the expected coverage as a function of budget $k$ on all datasets. Parameter $T = 5$, $\delta = 25$ and $\Delta T = 45$ are fixed.

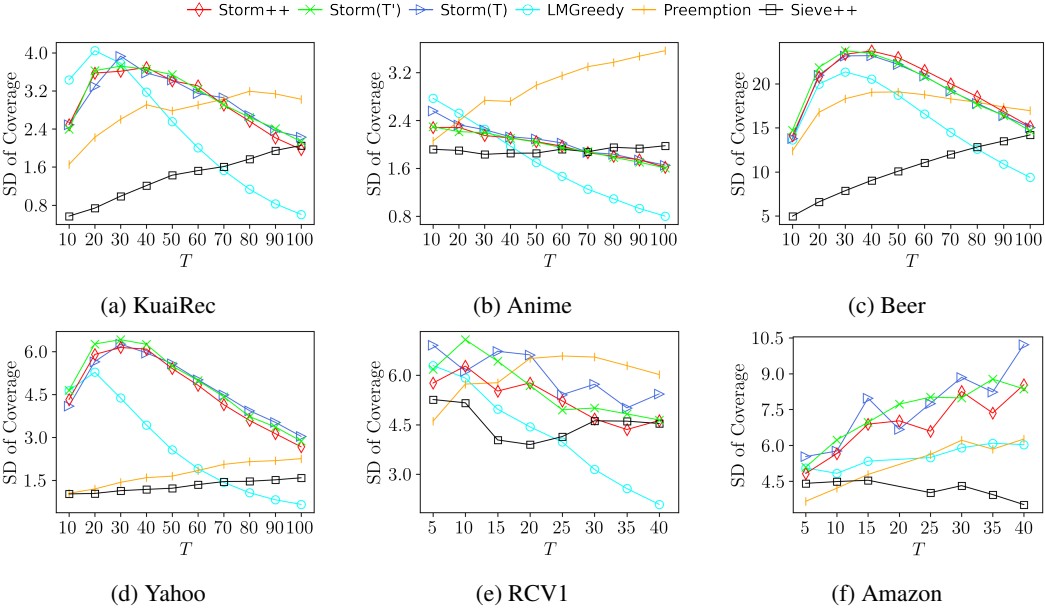

Figure 11: Empirical variation in standard deviation for the expected coverage as a function of the number of user visits $T$ on all datasets. Parameter $k = 10$, $\delta = 10$ and $\Delta T = 10$ are fixed.

