# OpenReview forum: "Streaming Stochastic Submodular Maximization with On-Demand User Requests"
_NeurIPS.cc/2025/Conference — NeurIPS 2025 poster_

### Official Review · Reviewer_whBa · 2025-06-27

**Clarity:** 3
**Significance:** 3
**Originality:** 3
**Rating:** 5
**Confidence:** 4

**Summary:**

This paper aims at a  streaming submodular maximization.  The context is that a user can visit a site and the site needs to present k distinct items to the users. These items will be consumed with a fixed probability (per item) which each item represents some topics. The goal is to design a streaming algorithm that maximizes the expected total topic coverage.

The authors show that this problem can be formulated as a submodular maximization subject to a matroid constraint under an ideal case:   the number of user visits is known in advance or linear-size memory in the stream length is available.

To do away from these assumptions, the authors proposed another streaming algorithm that achieves a competitive ratio of 1/8\delta, where \delta controls the approximation quality.

**Questions:**

Overall, it is a piece of work that introduces a good approximation algorithm, and it has performance guarantees.

I enjoy reading the paper.

**Ethical Concerns:**

["NO or VERY MINOR ethics concerns only"]

**Limitations:**

N.A.

**Paper Formatting Concerns:**

N.A.

**Quality:**

3

**Strengths And Weaknesses:**

The strength of this paper is in discovering the "connection" with the submodular maximization.

The authors also provide a cute approximation algorithm which can achieve 1/8\delta, competitive ratio.

The presentation, in particular, on Section 3, can be improved since there are a lot of claims that need formal proof but I guess it is due to the page limit that the authors didn't put the proof in the main text.

Overall, I enjoy reading the paper.

---

> ### Author Rebuttal · Authors · 2025-07-30
>
> Thank you for your comments! We will improve the presentation and clarity in the next version.

---

> ### Comment · Reviewer_whBa · 2025-08-01
>
> I read the review and I am ok with it. I will keep my score.

---

> > ### Comment · Reviewer_whBa · 2025-08-06
> >
> > Hope this paper can be accepted :-)

---

### Official Review · Reviewer_87n5 · 2025-07-01

**Clarity:** 3
**Significance:** 2
**Originality:** 2
**Rating:** 3
**Confidence:** 4

**Summary:**

This paper investigates a novel variant of the streaming submodular maximization problem. The authors consider a setting in which users may visit a news website at any time, and upon each visit, the website must display up to $k$ news items. Each news item covers a subset of topics and has a certain probability of being clicked by the user. The objective is to design a streaming algorithm that maximizes the expected total topic coverage. To address this, the authors draw a connection to submodular maximization under a matroid constraint.

**Questions:**

Please see weakness section for my concerns regarding both the problem setting and technical contribution.

**Ethical Concerns:**

["NO or VERY MINOR ethics concerns only"]

**Final Justification:**

My concern regarding the necessity of storing all news items remains.

**Limitations:**

Yes.

**Paper Formatting Concerns:**

No.

**Quality:**

2

**Strengths And Weaknesses:**

The paper introduces a new variant of the streaming submodular maximization problem and proposes memory- and time-efficient algorithms to address it. However, I have several concerns about both the problem setting and the technical contributions:

1. The paper uses news recommendation as a motivating example, focusing on the development of memory-efficient online algorithms that avoid storing all incoming news items. While this may be reasonable for serving a single user, real-world news platforms typically serve a large number of users. To provide near-optimal recommendations across all users, storing all news items may still be necessary. For instance, if the proposed algorithm is applied independently to each user, the union of the candidate sets across all users could approximate the full set of news items, especially given users' diverse preferences. This observation challenges the practicality of the memory savings claimed by the authors.

2. It is unclear what concrete advantages the proposed algorithms offer over existing methods such as LMGreedy. In particular, the memory requirement of their algorithm depends on $T'$, an upper bound on $T$ (the number of visits). In the worst case, $T'$ could be as large as $N$, the total number of news items. In such scenarios, the proposed method may offer little to no improvement over baseline approaches—unless it is shown to be robust to inaccurate estimates of $T$, which is not thoroughly addressed in the paper.

---

> ### Author Rebuttal · Authors · 2025-07-30
>
> We are thankful to the reviewer for their insightful comments.
>
> ### Q1 . The paper uses news recommendation as a motivating example, focusing on the development of memory efficient online algorithms that avoid storing all incoming news items. While this may be reasonable for serving a single user, real-world news platforms typically serve a large number of users. To provide near optimal recommendations across all users, storing all news items may still be necessary. For instance, if the proposed algorithm is applied independently to each user, the union of the candidate sets across all users could approximate the full set of news items, especially given users’ diverse preferences. This observation challenges the practicality of the memory savings claimed by the authors.
>
> Thank you for your great question!
>
> In the paper, we primarily consider memory usage for a single user and did not address the case of multiple users. It turns out that even with multiple users, **STORM** still uses less memory than **LMGreedy**.
>
> To compare their memory usage, let's assume that for both STORM and LMGreedy, memory usage consists of two components:
>
> 1. **Common storage**: Shared by all users, storing all incoming news documents from the stream.
> 2. **Per-user storage**: Allocated individually for each user, storing document identifiers and user-item click probabilities.
>
> The common storage is the same for both algorithms, so we only analyze the per-user storage.
> Let $N$ denote the length of the stream.
>
> ### STORM
>
> For each user, STORM requires memory $\mathcal{O}(T'k)$, since we store:
> - $T'k$ document identifiers, and
> - $T'k$ associated probabilities.
>
> ### LMGreedy
>
> For each user, LMGreedy requires memory $\mathcal{O}(N + Tk)$, because we store:
> - $Tk$ document identifiers, and
> - $N$ probabilities (one for each document in the stream).
>
> Assuming each document identifier and each probability takes one unit of memory, and given $m$ users, the total **per-user memory** usage is:
>
> - **STORM**: $\mathcal{O}(mT'k)$
> - **LMGreedy**: $\mathcal{O}(m(N + Tk))$
> - **STORM++**: $\mathcal{O}\left(\frac{m{T'}^2 k }{\delta}\right)$
>
> When $kT'<<N$, STORM becomes significantly more memory-efficient than LMGreedy. When
> $(k{T'}^2)/\delta<<N$，STORM++ is more memory-efficient than LMGreedy.
> In practice, $T$ (number of visits per day) is typically small, if we consider, e.g., the average number of user visits versus the number of new Reddit posts (or YouTube videos) each day.
>
> ### Q2. It is unclear what concrete advantages the proposed algorithms offer over existing methods such as LMGreedy. In particular, the memory requirement of their algorithm depends on $T'$, an upper bound on $T$ (the number of visits). In the worst case, $T'$ could be as large as $N$, the total number of news items. In such scenarios, the proposed method may offer little to no improvement over baseline approaches—unless it is shown to be robust to inaccurate estimates of , which is not thoroughly addressed in the paper.
>
> The key advantage of STORM++ over LMGreedy lies in its faster response time. STORM always maintains a set of $T'$ candidate solutions. Upon receiving a user request, STORM returns the best solution among these candidates. Building on this, STORM++ selects the best guess from $\frac{T'}{\delta}$ candidates and presents it to the user. This process incurs a maximum computational cost of $\mathcal{O}(T'k)$.
>
> In contrast, LMGreedy does not maintain intermediate results. Upon request, the algorithm iterates all existing items and greedily selects the top $k$ items. This procedure takes at least $\Omega(Nk)$ time.
>
> We assume that $T'$, an estimate of $T$, is reasonably accurate and $T'<<N$.
> In practice, $T$ (number of visits per day) is typically small, and $T$ can be inferred by directly asking the user or by analyzing their daily interaction patterns.

---

> > ### Comment · Reviewer_87n5 · 2025-08-07
> >
> > I would like to thank the authors for their thorough rebuttals. However, I still have some concerns regarding the practicality of their results. While the per-user storage requirements may vary across algorithms, they typically involve storing only item identifiers, which I believe incurs significantly less overhead than storing the original files. On the other hand, since all original files must still be maintained in a shared storage system, it is unclear how significant the savings in per-user storage actually are.

---

> > > ### Author Response · Authors · 2025-08-08
> > >
> > > We thank the reviewer for their thoughtful feedback.
> > >
> > > 1. **On storing original news content:**
> > >
> > > We note that for some applications, saving the original news is not necessary. For example, Google News recommends news from CNN or BBC, but Google does not save the original news; instead, it crawls news websites and only saves news index data.
> > >
> > > 2. **On per-user storage:**
> > >
> > > Indeed, different algorithms save item identifiers in the per-user storage. However, we want to stress that, under our stochastic setting, it is also necessary to save the user–item click probabilities because they are personalized. For LMGreedy, the per-user storage cannot be ignored, as applying LMGreedy leads to a space complexity of $O(m(N+Tk))$ for $m$ users. Note that, when $m$ is significantly large, this space usage can exceed the sizes of the original files.
> > >
> > > (1) Original file storage.
> > >
> > > If each item requires $a$ bytes, storing all $N$ items uses $O(Na)$ bytes.
> > >
> > > (2) Per-user probabilities and identifiers.
> > >
> > > Assuming each probability$\/$identifier needs $O(1)$ bytes, LMGreedy requires $O(N)$ bytes per user, and $O(mN)$ bytes in total.
> > >
> > > In practice, storing a short video takes 30MB $\sim$ 1GB, i.e., $a \in (10^7, 10^9)$. However, China’s short-video platforms have roughly $m \approx 10^{9}$ users. Thus, for LMGreedy, the per-user memory cost is substantial and cannot be ignored.
> > >
> > > 3. **Even without considering storage, STORM++ has merits:**
> > >
> > > If we ignore memory complexity, the criteria for evaluating different algorithms should include (i) how good the recommended items are in terms of the objective value, and (ii) how fast the response time is upon a user request.
> > >
> > > For criterion (i), our proposed STORM++ provides a theoretical guarantee, while the practical algorithms might not. Although the competitive ratio of STORM++ is worse than that of LMGreedy, we note that LMGreedy is also part of our theoretical contribution, as it is not straightforward that the greedy algorithm can be used to solve the given on-demand setting.
> > >
> > > For criterion (ii), since STORM++ always maintains high-quality results, it responds fast to users' queries.

---

> ### Author Response · Authors · 2025-08-05
>
> Thank you for your review. We’d appreciate it if you could share whether our response resolved your concerns, or if there’s anything else you'd like us to address. We’re happy to clarify further if needed.

---

### Official Review · Reviewer_XZhc · 2025-07-02

**Clarity:** 3
**Significance:** 3
**Originality:** 3
**Rating:** 5
**Confidence:** 4

**Summary:**

The paper introduces the steaming stochastic submodular maximization with on-demand requests (S3MOR) problem, motivated by online news recommendation systems. The goal is to select $T$ news sets (corresponding to $T$ user visits) that maximize the expected number of topics covered. This problme is further formalized as monotone submodular maximization problem under a matroid constraint.

The main contribution are three approximation algorithms:
- Given sufficient memory, a 1/2-approximation algorithm is proposed with $\Omega(N+kT)$ memory usage and  $O(NTk)$ running time.
- To improve the memory usage, a $\frac{1}{4(T'-T+1)}$-approxiamtion algorithm is proposed with $O(T'k)$ memory usage and  $O(NT'k)$ running time, where $T'$ is a given upper bound to the number of user accesses $T$.
- To further achieve a constant approximation ratio, a $1/(8\delta)$-approximation algorithm is proposed with $O(T'^2k/\delta)$ memory usage and  $O(NT'^2k/delta)$ running time.

The empirical evaluation shows the proposed algorithms consistently outperform the baselines regarding function value, runtime and memory.

**Questions:**

- Line 16: The range of $\delta$ should be provided. Otherwise, it could be misleading that the approximation ratio could be better than $1/8$. Also, add parenthesis as $1/(8\delta)$
- Table 1: On line 182, it is discussed that the memory usage of LMGreedy is $\Omega(N+kT)$. However, it is listed as $O(N+kT)$ in the table.
- Line 224: Does $f_{\cup_{t\in [T']} \mathcal A^t}(V_j)$ represents the marginal gain?

**Ethical Concerns:**

["NO or VERY MINOR ethics concerns only"]

**Final Justification:**

I have read the rebuttal and other reviews. I will keep my score.

**Limitations:**

Yes

**Quality:**

3

**Strengths And Weaknesses:**

Strengths
- The paper introduces an interesting online setting for streaming submodular maximization problems, capturing real-world challenges in online news recommendations. It models the goal of maximizing topic coverage while adapting to dynamic user visits, making it applicable to recommendation systems that aim to diversify content exposure.
- The authors propose three progressively refined approximation algorithms with different trade-offs between accuracy and memory efficiency.
- The paper includes extensive experiments on six real-world datasets, validating the effectiveness of the proposed methods. The results demonstrate that the algorithms outperform baselines in terms of objective value (topic coverage), runtime efficiency, and memory consumption.
- The paper is well-structured and clearly written, making it accessible to readers.
The logical flow from problem formulation to theoretical analysis and empirical evaluation enhances readability.

Weaknesses
- The probability $p_i$ that the user clicks an item $i$ is not clearly explained before the formal problem definition (Problem 2.2). It remains unclear whether click probabilities remain static or evolve based on user behavior over time.
- The paper does not discuss how to estimate the total number of user visits $T$. Under the streaming setting, the predefined $T'$ sets may become insufficient if users continue visiting beyond expectations. The experimental setup assumes prior knowledge of T, which may not be realistic. The author may need to either add a discussion on how to guess $T$ and implement it in the experiment or add an assumption to $T$.

---

> ### Author Rebuttal · Authors · 2025-07-30
>
> We are thankful for the reviewer's valuable feedback.
>
> ### Q1. It remains unclear whether click probabilities remain static or evolve based on user behavior over time.
>
> Please refer to our response to Reviewer 9b0Z (Question 1) for how we set probabilities.
>
> ### Q2. The paper does not discuss how to estimate the total number of user visits $T'$
>
> We assume $T$ represents the number of times a user visits the system per day, which is typically small. We could estimate $T$ by either directly asking the user or analyzing their behavior. Once we have an estimate, we can set $T'$ close enough to $T$, for example, $T' \approx 1.1T$.
>
> We always assume $T' > T$. This is because we want to pre-allocate a memory space for each user, and the memory depends on $T$. If we assume the user will visit $T'$ times, we set the memory to be $T'k$. If $T' < T$, any algorithm cannot perform well, as we explained in lines 192-199.
>
> ### Line 16
> Thank you for pointing this out. In line 147, we mention $\delta \in \mathbb{N}$. We will clarify in Theorem 3.2 that $\delta$ is a positive integer.
> ### Table 1
> Thank you for catching the typo. It should be $\Theta(N + kT)$ both in line 182 and in the table. We will revise it in the next version.
> ### Line 224
> Yes, this represents the marginal gain. We will clarify this in the next revision of the paper.

---

### Official Review · Reviewer_9boZ · 2025-07-03

**Clarity:** 3
**Significance:** 2
**Originality:** 2
**Rating:** 3
**Confidence:** 3

**Summary:**

This work introduces and studies a new problem in streaming submodular maximization, called Streaming Stochastic Submodular Maximization with On-demand Requests, or S3MOR in short. This problem is motivated by the setting of wanting to diversify the news feed of a user, where the news coverage goes up only if the user actually clicks on an item. Formally, there is a stream $\mathcal{V} = \{V_1,\dots,\}$ of news items, and each item $V_i$ covers a subset of topics from a given universe of topics $C = \{c_1,\dots,c_d\}$. The problem solver is given each news item tagged with a value $p_i$ that is the probability that the item will be clicked on. Further, at every time step $i$, a binary variable $\tau_i \in \{0,1\}$ is observed indicating whether or not the user of interest is online or not; if $\tau_i = 1$ then the solver must pick some at most $k$ items from $\{V_1,\dots,V_i\}$ to maximize the topic coverage for the user. The quality of the solvers algorithm is measured as the expected number of topics covered over the course of the stream.

After introducing this problem, the authors establish a baseline by showing that this can be reduced to submodular maximization under a partition matroid constraint, and so by storing the entire data stream and applying a batch algorithm, they achieve a $1/2$ approximation - they call this algorithm LMGreedy. It is known that it is not possible to achieve better than a $1/2$ approximation in the worst case setting for streaming submodular maximization, but the downside of LMGreedy is its high memory usage.

So as to address this issue, the authors construct memory efficient streaming algorithms. One technical issue they come across is that their first method (STORM) achieves a good approximation of $1/4$ only when the number of visits made by the user is known in advance, and degrades reciprocally with the gap between the known upper bound on the number of visits and the number of visits actually made. To address this, they adopt an adaptive approach where they initialize a sequence of guesses for the number of visits by the users and adapt to the sequence. This allows them to interpolate smoothly by achieving an approximation factor of $1/8\delta$ for any fixed parameter $\delta\in\mathbb{N}$, and achieve space complexity $O(T'^2k/\delta)$ and time complexity $O(NkT'^2/\delta)$, where $T'$ is the actual number of times the user visits.

Finally, the authors conduct some experiments demonstrating that their methods do indeed allow one to trade off coverage and space/time complexity effectively.

**Questions:**

1. Could you please address what I've written in the Weaknesses section above?
2. In your experiments, how did you generate your $p_i$ values to run your experiments?

**Ethical Concerns:**

["NO or VERY MINOR ethics concerns only"]

**Final Justification:**

I have kept my score as is (a 3, slightly below the acceptance threshold) - I think my biggest concern is the novelty and impact of this work. While this is a good work, I don't think it meets the bar for NeurIPS.

**Limitations:**

I think I've address any issues I have with limitations above in the Weaknesses section.

**Quality:**

3

**Strengths And Weaknesses:**

Strengths:
1. The paper is well-written and easy to understand.
2. The treatment of the problem is comprehensive with reasonable theoretical results and experimental corroboration.

Weaknesses:
1. I think this work is reasonably interesting but maybe not very novel. Based on the problem description and the technical description, although S3MOR seems reasonably posed there are some aspects of the problem description which needs more elaboration. Why is it reasonable to assume we have a clean probability tag for each item, and how would this perform under misspecification? How does this compare to other algorithms in the streaming submodular maximization literature which maybe don't exactly match this problem description but can perhaps still be applied?

---

> ### Author Rebuttal · Authors · 2025-07-30
>
> We are thankful to the reviewer for their feedback of our paper:
>
> ### Q1. Why is it reasonable to assume we have a clean probability tag for each item, and how would this perform under misspecification?
>
> Thank you for your question! Could you please clarify what you mean by a **"clean probability tag"?**
>
> 1. **If you mean why we can assume these probabilities are provided:**
>
>    These probabilities can be interpreted as the user's preference for certain items. Many machine learning and deep learning models are capable of estimating such preferences. We can treat these models as black boxes to generate the probability $p_i$ required by our algorithm.
>
>    For instance, the Probabilistic Matrix Factorization (PMF [1]) model approximates the user-item preference matrix as the product of two low-rank matrices—one for users and one for items. Additionally, Wang et al. [2] proposed a supervised machine learning model that learns personalized, contextualized, and real-time probabilities representing the likelihood that a user $i$ has intent $v$ for an item on the current recommendation page.
>
>    It is worth noting that our algorithm incorporates the scenario where multiple copies of the same news item may appear at different positions in the news stream.
>    Each copy of the news item can be assigned a different probability.
>    This corresponds to the case where the user may be presented with the same news item multiple times, and the user accesses it each time with different probabilities.
>
>
>
> 2. **If you mean why the probabilities for an item are static, but not, for instance, time-dependent:**
>
>
>
>    STORM++ cannot handle dynamic probabilities—i.e., probabilities that change over time.
>
>    Recall that STORM always prepares what to present to the users in advance in *partitions*, and decides on the fly whether to add the news item to each partition or not whenever the algorithm processes a new news item.
>    Whenever the user visits the system, the algorithm presents one partition to the user.
>
>    Hereby, as user visit times are uncertain, it is infeasible to predetermine probabilities for each partition when computing topic coverage. Furthermore, STORM selects the optimal partition at each request, making it even more challenging to preassign dynamic probabilities.
>    However, this is a very interesting future work direction, and we appreciate you for pointing it out.
>
>
> ### Q2. How does this compare to other algorithms in the streaming submodular maximization literature that may not exactly match this problem but could potentially be applied?
>
>
>
> Below, we list three different aspects that make our problem different.
> The major differences between our problem and other streaming submodular maximization problems arise from the fact that we incorporate an online setting, which are detailed in (b) and (c), and we provide more explanations in the following paragraphs.
>
>
> - **(a) Uncertainty of whether user accesses the recommended items:** Each copy of the news item is assigned a probability indicating the likelihood that the user will access it when presented to the user.
>
> - **(b) Users' visit times are unknown:** Users may access the system at any time, and the system does not know in advance when or how many times each user will visit.
>
> - **(c) The system provides fast response:** At each user's request, the system returns a result set of size at most $k$, and the system's response should be fast.
>
> In other words, the algorithm must always be prepared to produce results at any point during the processing of the stream. This requirement distinguishes our setting from traditional streaming submodular maximization problems (as in [3,4]), where the algorithm is only required to output a result at the end of the stream.
>
> In certain special cases, specifically (i) when the algorithm has access to all items at all times (as discussed in Section 2), or (ii) when the exact number of user visits $T$ is known in advance (as described in Section 3.1), we can reduce our problem to submodular maximization under matroid constraints. However, we emphasize that these reductions are *non-trivial* and require the construction of *new streams* for analytical purposes, as demonstrated in Appendix C of the supplementary material.
>
> In the general case, where only an upper bound on $T$ is known, such reductions no longer apply. In this setting, we must develop entirely new algorithmic strategies, as outlined in Section 3.2.
>
>
>
> ### Q3. In your experiments, how did you generate your $p_i$ values?
>
> We explain how we generate probabilities in Appendix D.1. For the four item-rating datasets, we estimate click probabilities $p_i$ by performing low-rank matrix factorization on the user-item rating matrix. This yields latent vectors $w_u$ (user) and $v_m$ (item), and their inner product $w_u^\top v_m$ approximates user $u$’s rating of item $m$. We then apply min-max normalization to these predicted ratings to obtain click probabilities. For the RCV1 and Amazon datasets, we sample $p_i$ uniformly at random from the range $[0, 0.2]$.
>
> We would like to mention that, in the literature, paper [5] employs a similar approach to generate probabilities.
>
>
>
>
> > [1] Mnih A, Salakhutdinov RR. Probabilistic matrix factorization. Advances in neural information processing systems. 2007;20.
>
> > [2] Wang Y, Banerjee C, Chucri S, Soldo F, Badam S, Chi EH, Chen M. Beyond item dissimilarities: Diversifying by intent in recommender systems. In Proceedings of the 31st ACM SIGKDD Conference on Knowledge Discovery and Data Mining V. 1 2025 Jul 20 (pp. 2672-2681).
>
> >[3] Badanidiyuru A, Mirzasoleiman B, Karbasi A, Krause A. Streaming submodular maximization: Massive data summarization on the fly. InProceedings of the 20th ACM SIGKDD international conference on Knowledge discovery and data mining 2014 Aug 24 (pp. 671-680).
>
> >[4] Chekuri C, Gupta S, Quanrud K. Streaming algorithms for submodular function maximization. In International Colloquium on Automata, Languages, and Programming 2015 Jun 20 (pp. 318-330). Berlin, Heidelberg: Springer Berlin Heidelberg.
>
> > [5] Wang, H., Tu, S., & Gionis, A. (2025, March). Sequential Diversification with Provable Guarantees. Proceedings of the 18th ACM International Conference on Web Search and Data Mining, pp. 345–353.

---

> > ### Comment · Reviewer_9boZ · 2025-08-06
> > **Response**
> >
> > Thank you for your detailed response! Regarding 'clean probability tags', your first interpretation addresses what I meant. After reading your response and your discussions with the other reviewers, I will keep my score as is. I think overall I still have concerns about the novelty and impact of this problem setting and solution.

---

> ### Author Response · Authors · 2025-08-05
>
> Thank you for your review. We’d appreciate it if you could share whether our response resolved your concerns, or if there’s anything else you'd like us to address. We’re happy to clarify further if needed.

---

### Note · Authors · 2025-08-12

We thank all reviewers for their valuable feedback and discussion.

Taking the opportunity to provide author final remarks, we would like to summarize the discussion during the rebuttal phase from our point of view and provide a list of revisions that we plan to incorporate in our paper.

The discussion focuses on two main points: (a) the validity of our problem setting, and (b) the practicality of our proposed **STORM++** algorithm.

(a) Regarding the problem setting, the reviewers comment on the rationality of the probabilistic framework and ask how we estimate the number of user visits, $T$ and $T'$.

We have addressed both questions. To improve clarity in the next version of the paper, we will further  emphasize the following points:

1. The probabilities indicate user–item preference or relevance and can be estimated from historical data using machine-learning models.
2. The proposed **STORM++** algorithm currently handles only static probabilities due to the on-demand setting.
3. We can estimate $T$ either by asking users when they join the system or by inferring it from their historical interaction data. We then set $T'$ close to $T$ within a certain range, for example, $T' \approx 1.1T $.
4. We will further emphasize that the assumption $T' >T $ corresponds to an **out-of-memory configuration**; any algorithm with less than $\Theta(Tk)$ memory for computing an output cannot achieve good results.

(b) Regarding the practicality of the proposed **STORM++** algorithm, we have discussed: (i) whether **STORM++** has better space complexity compared to the baseline, **LMGreedy**; and  (ii) the advantages of **STORM++** over **LMGreedy**.

Again, we have addressed both questions. To enhance understanding, we will incorporate the following points into the next version of the paper:

1. **Separating the concepts of common storage and individual storage.**
   - *Common storage* stores original files.
   - *Individual storage* is used by the recommender system to compute outputs for users.
   All algorithms share the same common storage, but **STORM++** achieves better memory efficiency than **LMGreedy** in terms of individual storage.
2. **Separating the concepts of computational time and response time.**
   The proposed **STORM++** offers faster response time than **LMGreedy** for user requests, as it always maintains a strong set of candidate solutions.

Lastly, we will correct typos and refine the writing for greater clarity in the next version.

---

### Decision · Program_Chairs · 2025-09-17

**Decision:**

Accept (poster)

**Comment:**

This paper investigates a novel variant of the streaming submodular maximization problem. The authors consider a setting in which users may visit a news website at any time, and upon each visit, the website must display up to k  news items. Each news item covers a subset of topics and has a certain probability of being clicked by the user. The objective is to design a streaming algorithm that maximizes the expected total topic coverage. To address this, the authors draw a connection to submodular maximization under a matroid constraint.

All reviewers agree that the paper is well-written and the treatment of the problem is adequately deep. Still there are some minor issue that limit in some way the impact of the paper: (i) it is not clear how the probabilities are provided in a recommendation system; (ii) it is limiting to think about single-user recommendations.

Anyway, the work's merits outweigh these issues, and the paper can foster more research addressing them. For this reason, we suggest acceptance.